# Competing climate feedbacks of ice sheet freshwater discharge in a warming world

Dawei Li [1,2,3] ✉, Robert M. DeConto [4], David Pollard [4,5] & Yongyun Hu [6]

Freshwater discharge from ice sheets induces surface atmospheric cooling and subsurface ocean warming, which are associated with negative and positive feedbacks respectively. However, uncertainties persist regarding these feedbacks' relative strength and combined effect. Here we assess associated feedbacks in a coupled ice sheet-climate model, and show that for the Antarctic Ice Sheet the positive feedback dominates in moderate future warming scenarios and in the early stage of ice sheet retreat, but is overwhelmed by the negative feedback in intensive warming scenarios when the West Antarctic Ice Sheet undergoes catastrophic collapse. The Atlantic Meridional Overturning Circulation is affected by freshwater discharge from both the Greenland and the Antarctic ice sheets and, as an interhemispheric teleconnection bridge, exacerbates the opposing ice sheet's retreat via the Bipolar Seesaw. These results highlight the crucial role of ice sheet-climate interactions via freshwater flux in future ice sheet retreat and associated sea-level rise.

Ice sheets wax and wane over the course of glacial-interglacial cycles, interacting with Earth's fluid (atmosphere and ocean) and solid (lithosphere and asthenosphere) shells in a variety of ways. These interactions include positive feedbacks that amplify the response to external forcings and negative feedbacks that inhibit or slow responses to perturbations. Competing negative and positive ice-climate feedbacks associated with ice sheet growth and retreat are responsible for the sawtooth signature of Pleistocene glacial-interglacial cycles[1,2]. Freshwater discharge from disintegrating ice sheets during the deglaciation phase of glacial-interglacial cycles could be on the order of 1 Sv ($10^6$ m$^3$ s$^{-1}$), equivalent to a rate of global-mean sea level rise (GMSLR) ~9 cm year$^{-1}$, and similar in magnitude to the present-day total global river discharge (~1.44 Sv[3]). Climate feedbacks associated with ice sheet freshwater flux (FWF)—here referring to discharge in both liquid (meltwater from ice surface and base) and solid (icebergs calved from ice shelves) forms—have been proposed as a trigger mechanism for abrupt climate change events[4]. In present-day climate conditions, warm, salty surface waters flow northward in the Atlantic

basin and are cooled in the high latitudes of the North Atlantic, forming deep waters that sink to depths and spread to the Southern Ocean, where they mix into the World Ocean[5]. This Atlantic Meridional Overturning Circulation (AMOC) transports heat across the equator to the North Atlantic and influences inter-hemispheric energy balance of the climate system. "Hosing" experiments, in which salinity fluxes are imposed on the ocean component of climate models to mimic ice sheet FWF, have demonstrated that ice sheet FWF stratifies the upper ocean and disturbs deep water formation, exerting global-scale climate impacts by changing the AMOC[5–9].

Ice sheet modeling under high-emission scenarios suggests the Antarctic Ice Sheet (AIS) alone may provide a peak FWF to the Southern Ocean exceeding 1 Sv in the next century[10], comparable to the Meltwater Pulse 1A of the Last Deglaciation[11]. Substantial climate impacts would be expected from ice sheet FWF on this scale, as evidenced by large climate disruptions during the Last Deglaciation coincident with ice sheet meltwater pulses, such as the Younger Dryas and the Antarctic Cold Reversal[12]. Nowadays ice sheet geometry and

[1]Key Laboratory of Polar Ecosystem and Climate Change, Ministry of Education, and School of Oceanography, Shanghai Jiao Tong University, Shanghai 200030, China. [2]Shanghai Frontiers Science Center of Polar Science, Shanghai Jiao Tong University, Shanghai 200030, China. [3]Key Laboratory for Polar Science, Polar Research Institute of China, Ministry of Natural Resources, Shanghai 200136, China. [4]Department of Earth, Geographic, and Climate Sciences, University of Massachusetts Amherst, Amherst, MA 01003, USA. [5]Earth and Environmental Systems Institute, Pennsylvania State University, University Park, PA 16802, USA. [6]Laboratory for Climate and Ocean-Atmosphere Studies, Department of Atmospheric and Oceanic Sciences, School of Physics, Peking University, 100871 Beijing, China. ✉e-mail: lidavvei@sjtu.edu.cn

discharge locations of ice sheet meltwater are very different from those over past glacial-interglacial cycles. Climate perturbations caused by past meltwater pulses, therefore, should not be regarded as direct analogs of what would happen in intensive future warming scenarios, prompting the need for modeling the climate effects of future ice sheet melt. Experiments using different numerical models reveal robust climate responses to ice sheet melt. Freshening of the upper ocean by meltwater enhances stratification and leads to suppressed vertical mixing and heat exchange, cooling the atmosphere and ocean surface and warming the subsurface ocean[13–17]. These responses, in turn, are expected to influence ice sheet melting and retreat. Surface cooling reduces ice surface melting and meltwater-induced hydrofracturing and calving of ice shelves−a negative feedback, while subsurface warming enhances basal melting of ice shelves −a positive feedback. Which feedback would prevail during the retreat of the AIS under anthropogenic warming, however, remains unclear. Contradicting conclusions have been drawn out of simulations using offline-coupled ice sheet models (discussed further in the methods section) and atmosphere-ocean climate models, suggesting either the positive feedback associated with meltwater-induced subsurface warming could lead to further ice loss and aggravated sea level rise[13,15,18], or the negative feedback from surface cooling could delay the progress of anthropogenic warming and its detrimental consequences, including the purported collapse of the marine-based portions of the AIS[17,19]. Although various Earth system models with built-in ice sheet components are under active development, e.g., the UKESM[20] and the E3SM[21], they have not been used for studying centennial-scale ice sheet-climate feedbacks. Since ice sheet-climate interactions and associated feedbacks cannot be reliably simulated in the offline coupling framework with prescribed climate/ice sheet boundary conditions that preclude time-evolving interactions between the ice sheet, atmosphere, and ocean, the net effect of ice sheet FWF-climate interactions on future retreat of ice sheets remains unknown.

Here we provide numerical simulations with a three-dimensional ice sheet model (ISM) quasi-synchronously coupled to a climate model of reduced complexity with a coupling time step of 1 year. The ISM is a 3-D dynamic-thermodynamic model that simulates ice sheets' surface and basal mass balance using bias-corrected climate fields from the climate model, as well as processes including basal sliding and bedrock deformation[22]. The climate model is an Earth system model of intermediate complexity (EMIC) that includes a three-dimensional ocean model, a land model, and a two-dimensional energy-moisture balance model for the atmosphere[23]. Using a reduced-complexity climate model enables carrying out a large set of multi-century scale simulations with different combinations of model configurations, climate sensitivities, emission scenarios, and initial conditions. Limitations of such model choices will be discussed in the Discussion section. The climate model's output drives ice sheets' mass balance and changes, while the ISM feeds back its simulated ice surface elevation and ice sheet FWF to the climate model. We focus on the interactions between ice sheet FWF and the climate for both the AIS and the Greenland Ice Sheet (GIS) in historical-future climate scenarios specified by six Shared Socioeconomic Pathways (SSPs). To account for uncertainty in the equilibrium climate sensitivity (ECS) of Earth's climate system, we scale the atmospheric $CO_2$ concentration ($pCO_2$) in historical and future scenarios up/down accordingly to emulate ECS higher/lower than the climate model's intrinsic ECS. Four configurations of ice sheet-climate coupling are devised: (1) Ice sheet FWF passed to the ocean is kept constant at the preindustrial level for both ice sheets, though ice sheets respond to changes in the climate and their potential contribution to sea level is recorded, (2, 3) FWF from one ice sheet interacts with the climate (termed "interactive FWF" hereafter) while the other remains fixed, and (4) FWF is fully interactive from both ice sheets. In all configurations ice sheets can interact with the climate by feeding back their surface elevation to the climate model as a surface boundary condition. With ensemble runs to suppress noise due to internal variability, these configurations enable isolating the effect of FWF-climate interactions for each or both ice sheets.

## Results

### Scenario-dependence of ice sheet freshwater-climate feedbacks

Substantial uncertainty persists with respect to the magnitude of future global warming, due not only to the uncertainty in future greenhouse gas emission, but also to the sensitivity of Earth's climate in response to greenhouse gas forcing, which can be quantified by the metric ECS—the eventual rise in global-mean surface air temperature (GMSAT) in response to doubling $pCO_2$. Models participating in the latest generation (phase 6) of Climate Model Intercomparison Project (CMIP6) display a range of ECS from 1.8 to 5.6 °C[24]. Under the same emission scenario, the large spread in ECS leads to very different changes in GMSAT across CMIP6 climate models. Inter-model differences in projected polar temperatures are even greater due to polar amplification, which would result in divergent future trajectories of polar ice sheets when climate model outputs are used to drive ice sheet models[25]. To study the climate feedbacks of ice sheet freshwater flux under a warming climate, it is necessary to consider a range of ECS in combination with representative warming scenarios, enveloped by optimistic cases with a low ECS and strongly mitigated emissions, and worse cases with a high ECS and intensive emissions. UVic-ESCM, the climate model used in our study, has an ECS of 3.4 °C[26], close to the best estimate of 3.0 °C assessed by IPCC-AR6[27]. The model, however, displays a -30% weaker polar amplification compared with more sophisticated climate models (Supplementary Fig. 4), thereby modeling less polar warming than a typical CMIP6 model with the same ECS ("Methods"). Given the same increase in $pCO_2$, UVic-ESCM with an emulated ECS of 4.0 °C would simulate polar warmings of roughly the same magnitude as typical CMIP6 models with an ECS around 3.0 °C would do. Considering these factors, we scale the $CO_2$ levels in historical-future scenarios to emulate three representative ECS (3.0 °C, 4.0 °C, and 5.6 °C), which produce roughly the same changes in polar temperatures as CMIP6 models with ECS of 2 °C, 3 °C, and 4 °C, respectively ("Methods"). Driven by these $CO_2$ trajectories, the coupled ice sheet-climate model (with interactive FWF from both ice sheets) shows that the peak ice sheet freshwater flux within the next few centuries varies by an order of magnitude between these scenarios (Fig. 1a, b). The warmest future scenario considered here shows a peak FWF of 0.37 Sv for the GIS, and -1.1 Sv for the AIS, a nearly total loss of the GIS by 2500, a collapse of the WAIS peaking around 2300 and an AIS contribution to GMSLR exceeding 10 m by 2500 (Fig. 1). In this warmest scenario, collapse of the WAIS is initiated around 2100, and is nearly concluded by 2400, with a contribution to sea level rise of more than 4 m over three centuries. The Atlantic Meridional Overturning Circulation (AMOC) weakens in all scenarios until the mid-twenty-first century, then diverges with continuous decline in the warmest scenarios but recovery and overshoot in scenarios with less warming.

A group of simulations are carried out with fixed ice sheet FWF, i.e., ice sheet FWF received by the climate model is kept same as the long-term mean FWF simulated by the coupled ISM-climate model in pre-industrial conditions, but potential ice sheet sea level contribution is recorded. By comparing the outputs of simulations with fixed ice sheet FWF versus those with the same $CO_2$ trajectory but with interactive ice sheet FWF, we can evaluate the effect of ice sheet FWF-climate interactions. Increased ice sheet FWF generally reduces future global warming in all scenarios, with the largest reductions associated with higher ice sheet FWF (Fig. 1d). Weakening of the AMOC is also more substantial in scenarios with more intensive ice sheet melt (Fig. 1f). The GIS and the AIS, interestingly, display contrasting responses to ice sheet FWF-climate interactions (Fig. 1h, j). The GIS retreats slower in most scenarios with interactive FWF, indicating an

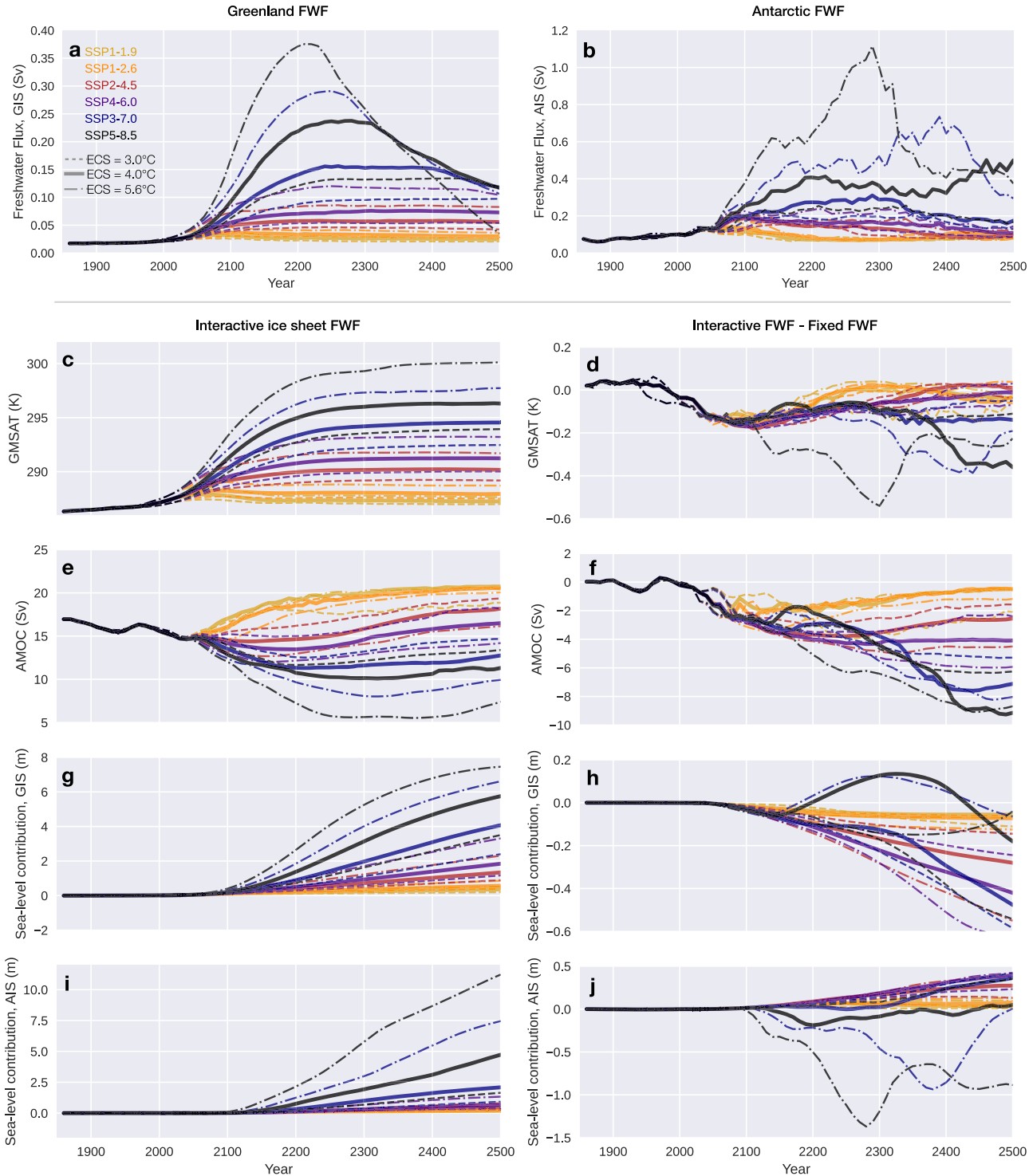

**Fig. 1 | Time series of selected variables simulated by the coupled ice sheet-climate model in a variety of scenarios. a** Freshwater flux (FWF) from the Greenland Ice Sheet (GIS) in the FWAG configuration (FWF of both ice sheets interacts with the climate). **b** Same as (**a**) but for the Antarctic Ice Sheet (AIS). **c**, **e**, **g**, **i** Results from the FWAG configuration. **d**, **f**, **h**, **j** Difference in results between the FWAG and the FWC (constant ice sheet FWF) configurations, showing the effects of interactive ice sheet FWF. Atmospheric $CO_2$ concentration is specified according to the historical (1850–2014) and six SSP scenarios (2015–2500), which are color-coded. Simulations with ECS of 3.0 °C, 4.0 °C, and 5.6 °C are shown in dashed, solid, and dash-dot lines, respectively.

overall negative feedback (Fig. 1h). This is consistent with the more pronounced and prolonged weakening of the AMOC in simulations with interactive FWF (Fig. 1f), resulting in subdued meridional ocean heat transport into the North Atlantic and a relative cooling there (Supplementary Fig. 8). Exceptions are seen in scenarios with the largest warmings, but these are likely due to trans-hemispheric influence

from the Antarctic FWF, as discussed later. The AIS, in contrast, is dominated by the positive feedback, with the ice sheet retreating faster when FWF interacts with the climate in most scenarios, except those with the largest warmings and consequently the largest ice sheet FWF (Fig. 1j). Both positive and negative feedbacks of ice sheet FWF-climate interactions are at play for the AIS, but the negative feedback

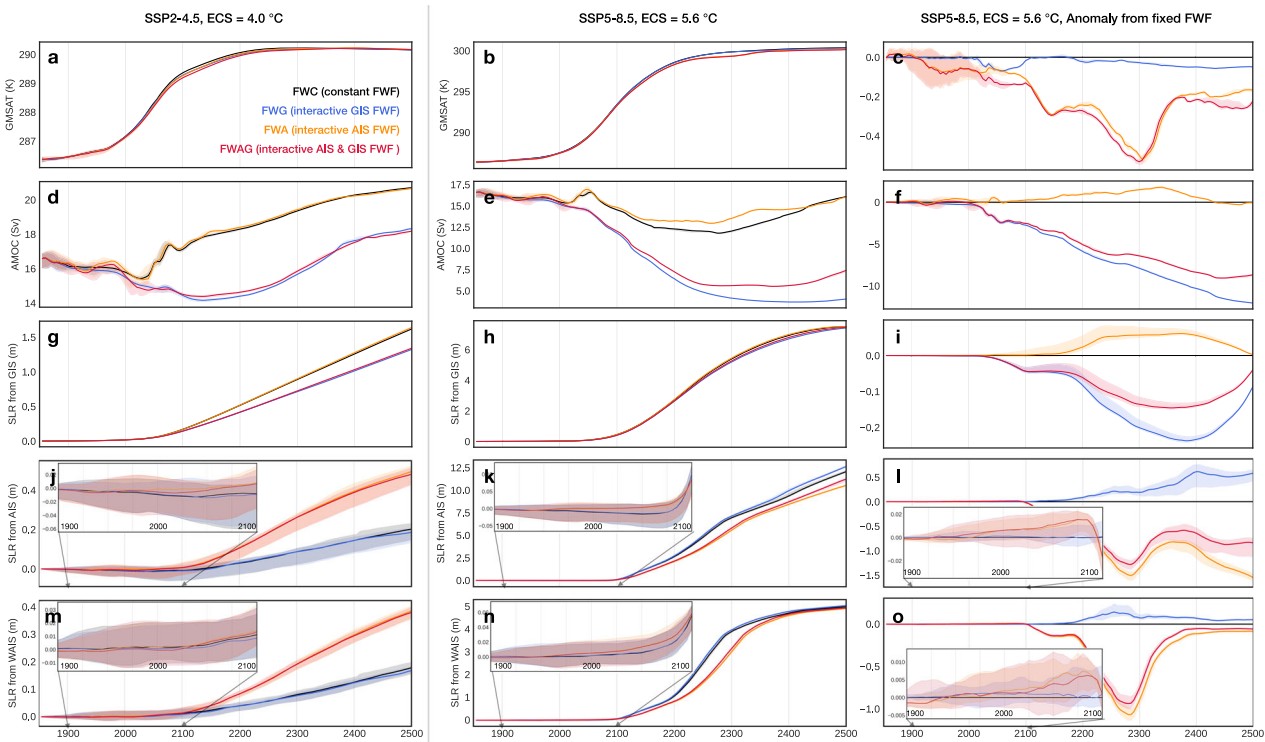

**Fig. 2 | Time series of selected variables from 10-member ensemble simulations with the coupled ice sheet-climate model in two CO$_2$ scenarios and four freshwater flux (FWF) configurations.** Top to bottom rows show global-mean surface air temperature (GMSAT, **a**–**c**), intensity of the Atlantic Meridional Over-turning Circulation (AMOC, **d**–**f**), sea level rise (SLR) contributed by the Greenland Ice Sheet (GIS) (**g**–**i**), by the Antarctic Ice Sheet (AIS) (**j**–**l**), and by the West Antarctic Ice Sheet (WAIS) (**m**–**o**). Left column **a**, **d**, **g**, **j**, **m**, CO$_2$ follows the SSP2-4.5 scenario with an Equilibrium Climate Sensitivity (ECS) of 4.0 °C. Middle column **b**, **e**, **h**, **k**, **n**, a more intensive warming scenario where CO$_2$ follows SSP5-8.5 with an ECS of 5.6 °C. Right column **c**, **f**, **i**, **l**, **o**, same as middle column but showing the anomaly relative to the FWC configuration (constant ice sheet FWF). In each panel, solid lines show the 10-member mean, and shadings show the spread between ensemble members. Insets in two bottom rows zoom-in to the period 1900–2100.

strengthens faster with increasing FWF and prevails in the warmer scenarios.

## Competition between positive and negative feedbacks in Antarctica

To obtain reliable signals, we carry out 10 ensemble simulations in each model configuration for two future warming scenarios, one representing a moderate warming with the $p$CO$_2$ of SSP2-4.5 scaled to emulate an ECS of 4.0 °C, the other an intensive warming with the $p$CO$_2$ of SSP5-8.5 scaled to emulate an ECS of 5.6 °C ("Methods"). Configurations with interactive Greenland FWF exhibit stronger and more prolonged weakening of the AMOC. In the intensive warming scenario, interactive Greenland FWF suppresses the AMOC throughout the simulations, preventing its recovery and overshoot (Fig. 2e, f). Interactive Greenland FWF has a relatively small influence on the GMSAT, exhibiting a depression of ~0.1 °C in the warmest scenario, substantially weaker than that from the AIS FWF around the time of WAIS collapse (~0.5 °C, Fig. 2c). The Greenland FWF-induced surface cooling of ~1 °C is concentrated in the North Atlantic to the south of Greenland accompanied by weaker warming elsewhere (Fig. 3), lessening its global-mean impact.

Ice sheet-climate feedbacks associated with Antarctic FWF are more complex than the Greenland case, involving competing negative and positive feedbacks. The sign of the net feedback is positive in the moderate warming scenario, where interactive Antarctic FWF accelerates ice loss throughout the simulation, doubling GMSLR attributable to Antarctica by 2500 (Fig. 2j). In the intensive warming scenario, although interactive Antarctic FWF accelerates ice loss in early stages of warming before 2100, thereafter the negative feedback comes to dominance, substantially delaying and slowing down the WAIS

collapse (Fig. 2k). In mid 2200s, when the WAIS undergoes runaway retreat, interactive Antarctic FWF results in a WAIS contribution to GMSLR ~30% less than that from simulations without interactive Antarctic FWF. After 2400 in the intensive warming scenario, when the WAIS has collapsed in all four configurations, interactive Antarctic FWF remains effective in reducing the rate of ice loss from the East Antarctic Ice Sheet (Fig. 2l).

Elevated Antarctic FWF from the retreating ice sheet enhances stratification of the upper ocean, thereby suppressing heat exchange between the cold Antarctic atmosphere and the relatively warm subsurface ocean, resulting in surface cooling and subsurface warming (Fig. 3b, d). Additional surface cooling is generated by latent heat absorption by melting icebergs, calved from ice shelves constituting the solid fraction of FWF. Though icebergs are not explicitly simulated in our model and are treated as imposed sea ice flux ("Methods"), the associated energy budget remains valid. In year 2300 of simulations following the SSP5-8.5 scenario with an emulated ECS of 5.6 °C, when the WAIS is under a runaway retreat, the surface cooling effect of ice sheet FWF exceeds 3 °C over West Antarctica, while the subsurface warming approaches 3 °C in the Weddell Sea (Fig. 3d). The fresher surface ocean, provided with less upwelling heat from the subsurface, is subject to more sea ice formation (Fig. 3d). This expansion of sea ice introduces additional surface cooling via the ice-albedo feedback. Cooling of the ocean surface and the near-surface air would alleviate surface melting of the ice sheet and its ice shelves, constituting a negative feedback[19]. On the other hand, enhanced basal melting due to the subsurface oceanic warming thins ice shelves and speeds their flow, providing less buttressing to the grounded ice upstream and exacerbating ice sheet loss in a positive feedback. This positive feedback operates at the base of ice shelves, so its strength declines as the

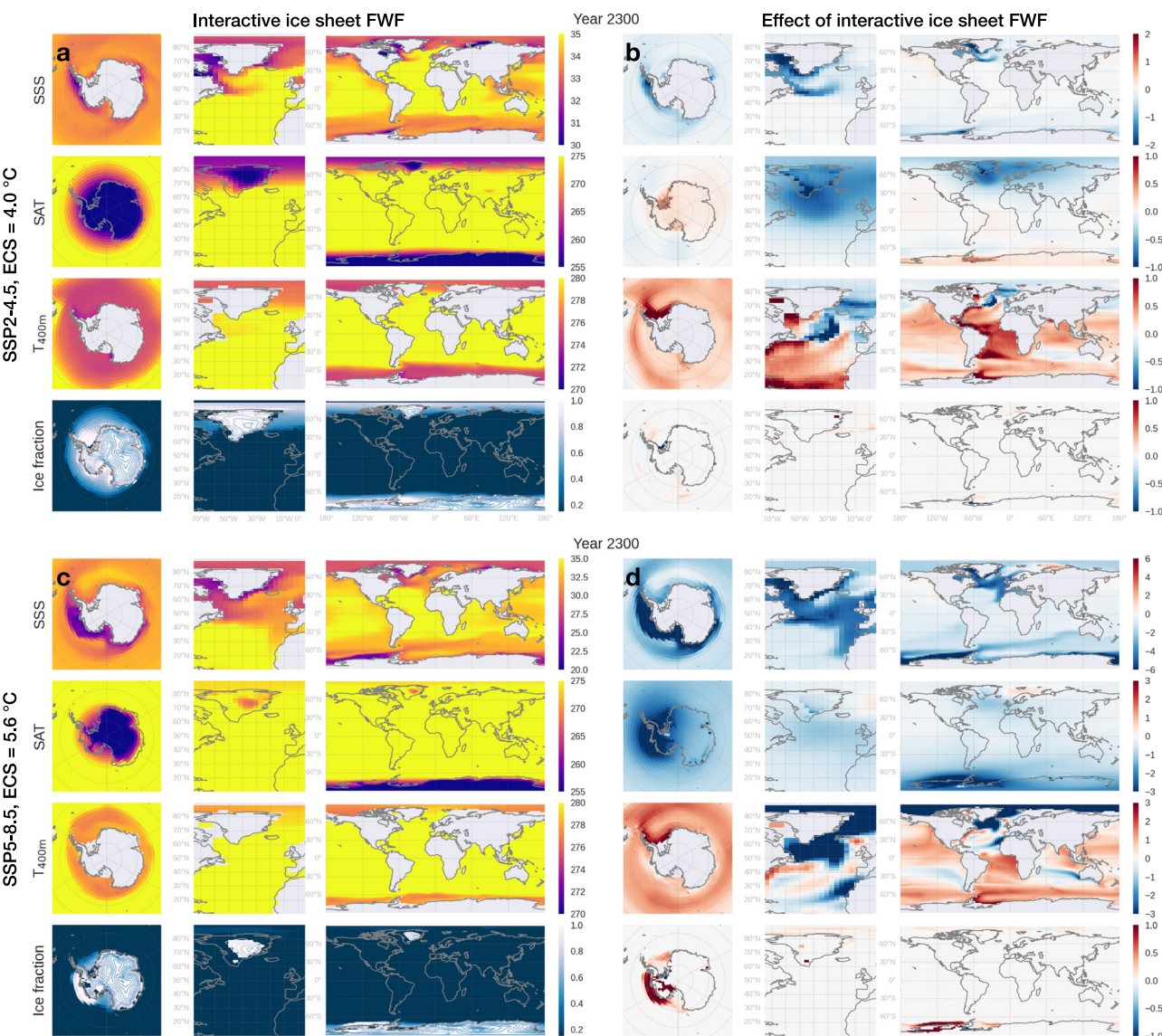

**Fig. 3 | Snapshots of selected variables in year 2300 simulated by the coupled ice sheet-climate model with interactive freshwater flux (FWF) from both ice sheets in two $CO_2$ scenarios.** Rows from top to bottom in each panel show sea surface salinity (SSS), surface air temperature (SAT), 400 m depth ocean temperatures ($T_{400\,m}$), and ice fraction (blue contours show ice surface elevation with an interval of 200 m). **a** $CO_2$ follows SSP2-4.5 with an Equilibrium Climate Sensitivity (ECS) of 4.0 °C. **b** Same as (**a**) but showing the difference between simulations with interactive FWF from both ice sheets (FWAG) and those with constant ice sheet FWF (FWC). **c**, **d** Same as (**a**, **b**) but for a more intensive warming scenario where $CO_2$ follows SSP5-8.5 with an ECS of 5.6 °C. Yearly evolution of these maps from 1850 to 2500 under historical-SSP2-4.5 (ECS = 4.0 °C) and historical-SSP5-8.5 (ECS = 5.6 °C) are available in Supplementary Movies 1 and 2, respectively.

polar climate warming proceeds and ice shelves shrink in size. The negative surface cooling feedback, in contrast, is not solely dependent on the presence of ice shelves. This difference in locality between the positive and negative feedbacks may explain the sign-reversal of the net feedback in the intensive warming scenario after early 2100s (Fig. 2n). In the moderate warming scenario, however, Antarctic ice shelf area does not dwindle as substantially, thereby maintaining a net positive feedback.

The effect of ice sheet FWF-climate feedbacks on the AIS' mass loss can be quantified by a feedback factor $\gamma$, which is defined as one minus the ratio between the AIS' mass loss rates in simulations with interactive ice sheet FWF and those in simulations with fixed ice sheet FWF ("Methods"). A positive feedback factor $\gamma$ indicates that ice sheet FWF-climate feedbacks accelerate the AIS' mass loss, and vice versa. Figure 4 presents the feedback factor $\gamma$ as a function of the rate of ice loss from Antarctica (in Sv, freshwater flux equivalent). $\gamma$ is positive for low ice loss rates and transitions to negative values around a threshold

of 0.2 Sv. The transition of the net feedback from positive to negative is consistent with Fig. 2. The peak amplitude of $\gamma$ exceeds 0.5 for both the positive and the negative feedback regimes, indicating moderately strong feedbacks for ice sheet FWF-climate interactions. The strength of the negative feedback decreases as the ice loss rate exceeds ~0.4 Sv, corresponding to the stage of the WAIS collapse. In this stage, ice sheet instability mechanisms are at work, and the rate of ice loss is more strongly affected by ice sheet dynamics than atmospheric and oceanic thermal forcings, which may explain the decreasing feedback strength and irregularities at very large ice loss rates.

**Inter-hemispheric ice sheet interactions via the AMOC**

Climate feedbacks associated with ice sheet FWF are not restricted to the respective ice sheet and its local climate, but display global influences. Compared with the simulation without interactive ice sheet FWF, in model configurations where FWF from either ice sheet is interactive, the other ice sheet's mass loss is accelerated in the

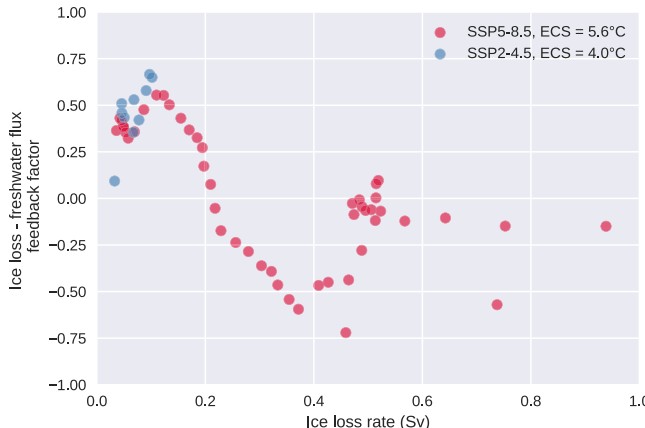

**Fig. 4 | Ice loss-freshwater flux feedback factor as a function of ice loss rate from the Antarctic Ice Sheet.** Blue markers are based on simulations under the historical-SSP2-4.5 scenario with an Equilibrium Climate Sensitivity (ECS) of 4.0 °C, while red markers are based on simulations under the historical-SSP5-8.5 scenario with an ECS of 5.6 °C. Definition of the feedback factor and its calculation are described in "Methods".

intensive warming scenario (Fig. 2i, l). The presence of interactive Greenland FWF increases the Antarctic contribution to GMSLR by ~0.5 m around year 2500, compared to simulations without interactive Greenland FWF (Fig. 2l). The influence of Antarctic FWF on the pace of GIS retreat, although smaller in magnitude (~0.1 m through 2300s), is nevertheless a robust feature that emerges unambiguously in the ensemble simulations (Fig. 2i). These results suggest melting of one ice sheet amplifies other ice sheet's mass loss. Note that by design, the numerical experiments keep sea level fixed, so the GIS and the AIS do not interact via their contribution to sea level changes as in the real world[28]. The trans-hemispheric impacts of ice sheet FWF shown here are likely dominated by influences on the AMOC.

Greenland meltwater injects freshwater into the region of North Atlantic Deep Water (NADW) formation, thereby weakening the AMOC and reducing the trans-hemispheric oceanic heat transport from the Southern Hemisphere (SH) to the Northern Hemisphere (NH). This causes a North Atlantic cooling accompanied by a SH warming, which increases Antarctic ice loss (Supplementary Figs. 10, 11, 13). In contrast, Antarctic meltwater freshens the source region of the Antarctic Intermediate Water (AAIW), which normally stays on top of the denser NADW. This reduces the density of AAIW, resulting in a stronger inflow of NADW into the SH and strengthening the AMOC (Supplementary Figs. 14–16 and Supplementary Movie 3), which warms the North Atlantic and enhances ice loss from the GIS (Supplementary Figs. 9, 11, 12). This inter-hemispheric link between high-latitude NH and SH climates, termed the "bipolar seesaw"[29], was proposed as a potential mechanism for the concurrence of the Bølling-Allerød warm interval in the NH and the Antarctic Cold Reversal in the SH during the Last Deglaciation[11,30,31].

## Discussion

Contradictory conclusions have been given in previous studies regarding the effect of ice sheet-climate interaction on the retreat of the AIS, with either a net positive feedback that accelerates its ice loss[15] (G19 hereafter), or a net negative feedback that ameliorates its decline[17,19] (S20 and D21 hereafter). These studies, however, investigate the feedbacks in integrated ice sheet-climate systems with similar offline or asynchronous coupling strategies, which cannot resolve time-evolving interactions between the ice sheet and climate. Here, we use a coupling time step of 1 year to enable near-synchronous coupling between the two models and better representation of their

interactions, and carry out an ensemble of simulations that allow for more robust differentiation of configurations with or without interactive ice sheet FWF. Our study considers a variety of future warming scenarios generated from a combination of six $CO_2$ emission pathways with three ECSs and demonstrates a strong dependence of the feedback's sign on the warming scenario. Concerning how the climate responses to ice sheet FWF feedback on ice sheet retreat, we find that previous conclusions are not necessarily conflicting and can be reconciled. In S20 and D21, the studies that show a net negative feedback, the Antarctic FWF were provided from ice sheet simulations forced by CCSM4, a climate model with an ECS of ~4.0 °C[32], roughly two times higher than that of the model used in G19 (ECS = 1.9 °C[33]). Simulations in S20 and D21 proceed to 2300, versus 2100 in G19, allowing a more thorough response of the AIS, hence more substantial retreat. The ice sheet model of S20 and D21 parameterizes hydro-fracturing and ice cliff failure—two key processes for the Marine Ice Cliff Instability (MICI) mechanism[34], which facilitate disintegration of ice shelves in short order under intensive warming scenarios, leaving less ice shelf basal area to melt. Introducing MICI processes also increases the sensitivity of ice loss to atmospheric warming (via meltwater production) and may strengthen the negative feedback. In our coupled simulations, turning off ice cliff failure in the ISM indeed reduces the strength of negative feedback in intense warming scenarios (Supplementary Fig. 17). These factors result in faster ice loss from the AIS (especially from its ice shelves) in S20 and D21 than G19, thus emphasizing the dominance of the negative atmospheric cooling feedback.

However, it is important to acknowledge the limitations of this study, and that ice sheet-climate interactions may be strongly model-dependent. The coupled ice sheet-climate model developed for this study was designed as a stopgap until more sophisticated Earth system models with fully integrated ice sheet components become mainstream. It provides a benchmark that can be used for comparison in subsequent studies using more comprehensive models. Compared with state-of-the-art climate models, the high throughput of the UVic-ESCM comes at the expense of reduced spatial resolution and model complexity, especially for its atmosphere model. Key aspects and processes of the modeling toolkit that need further improvement include but are not limited to:

- Spatial resolutions: the coupled ice sheet-climate model represents the large-scale processes controlling the feedbacks, but not the small-scale ones. Considerably higher spatial resolutions—in both ocean and atmosphere modules as well as in horizontal and vertical directions—are required to model processes vital for ice shelf basal melting, e.g., incursion of CDW onto continental shelves[35], and precipitation at ice sheet margins with steep topography. The response to ice sheet meltwater can strongly depend on the ocean model's spatial resolution and its parameterization schemes for mesoscale eddies and submesoscale eddy restratification. For instance, Antarctic meltwater is more efficiently trapped on the shelf in models with a better resolved and stronger Antarctic Slope Current (ASC), which produces subsurface cooling and suppresses further melt; In contrast, in models with a diffuse ASC, ice sheet meltwater more readily escapes to the open ocean, producing strong subsurface warming that accelerates further melt at the base of ice shelves[36].

- Ice shelf cavity circulations: the ISM used in this study assumes a simple quadratic relationship between ice shelf basal melt rates and 400 m depth ocean temperatures of nearby ocean cells, without explicitly modeling ocean circulations within the cavity beneath the ice shelf. Intensive basal melting of ice shelves of the Amundsen Sea can induce an overturning circulation in the ice cavity and an inflow of warm water into the cavity, which pumps heat from the deep ocean toward the ocean surface, melting sea ice near the ice sheet margins[37,38].

- Representation of icebergs: if the MICI mechanism is triggered by intensive warming, collapse of ice shelves and tall, mechanically unstable ice cliffs would release the bulk of FWF in the solid form— i.e., icebergs, which is not explicitly modeled in our study but is treated as added sea ice. Icebergs transport and release freshwater along their tracks from coastal Antarctica to warmer oceans, shifting the regions of freshwater injection equatorward[16]. In addition, most of the large tabular Antarctic icebergs are trapped in counter-clockwise currents along the coast for years before entering the "iceberg alley" of the Weddell Sea and drifting to lower latitudes[39]. This may shift the deposition of icebergs' freshwater flux westward off the calving sites and lead to more freshwater flux into the Indian Ocean. Iceberg tracks cannot be accurately modeled with the coarse-resolution velocity field from our model, but ocean models with built-in iceberg modules are under activate development[40,41].
- Depth of freshwater injection: the coupling scheme of this study injects ice sheet FWF to the top layer of the ocean model, while in the real world basal meltwater is injected at depth by both ice shelves and icebergs. This modeling choice is due to the lack of explicitly modeled ice shelf cavities and icebergs. Strengthening of ice shelf cavity circulations due to intensified basal melting may deepen the mixed layer around Antarctica, in contrast to iceberg meltwater, which is shown to enhance stratification[42]. Adding iceberg meltwater at depth was found to increase the magnitude of subsurface warming and sea ice trends[43]. This complexity could affect the positive feedback identified in this and previous studies.

Ice sheet FWF-climate interactions are currently not accounted for in most comprehensive climate models, compromising model-based future projections of ice sheet retreat and associated sea level rise. To address this need, a number of climate models equipped with fully interactive ice sheet components are under development[20,21]. Next-generation models, based on results shown in this study, are expected to project an overall faster retreat of the AIS and associated sea level rise under anthropogenic warming unless/until a catastrophic collapse of the WAIS is initiated. Using high-resolution, dynamically comprehensive models to investigate ice sheet-climate feedbacks is a challenging task, in part because the coupled system takes millennia to equilibrate due to the long timescales of the deep ocean and ice sheet. Returning to the question as to which feedback dominates, however, recognizing the limitations of this study, more advanced and comprehensive modeling tools may tip the scale in either feedback's favor. Mechanisms revealed by this coupled ISM-CM model study, including the dependence of ice sheet-climate feedbacks on the pace of warming, and the transition of the net feedback from positive to negative as ice shelves are lost, are potentially important for the sensitivity of ice sheets to climatic warming. They should be investigated more fully in coupled ice sheet-climate models that can better resolve the aforementioned processes that cannot be reliably represented by the model used in this study.

## Methods

### Ice sheet model

The ice sheet component of the coupled ice sheet-climate model is PSUICE3D[22], a continental-scale ice sheet model (ISM) using a hybrid approach for ice flow dynamics, i.e., the shallow ice approximation (SIA) for the ice sheet flow and the shelfy-stream approximation (SSA) for the ice shelf flow. These two flow regimes are heuristically fused by an imposed mass flux condition across the grounding line[22,44]. With the hybrid ice dynamics the ISM captures grounding line migration while running on relatively coarse-resolution grids (e.g., 10/20 km)[19], allowing runaway retreat to arise naturally for a marine ice sheet resting on a reverse-sloping bed, a mechanism termed the "Marine Ice Sheet Instability" (MISI). Bedrock deformation is calculated using an Elastic

Lithosphere/Relaxed Asthenosphere (ELRA) model, in which the weight of the ice sheet produces an elastic lithospheric flexure and a local asthenospheric relaxation toward isostatic equilibrium. As sub-ice shelf cavities are not resolved in most CMIP6 models and ocean reanalysis datasets, for ISM grid cells occupied (fully or partially) by ice shelves, 400 m ocean temperature of the nearest climate model grid cell is used for calculating ice shelf basal melt rates. Climate fields from the climate model's grid are bi-linearly interpolated to the finer ISM grid (20 km for Antarctica and 10 km for Greenland). A simple lapse-rate correction is applied to the interpolated surface air temperature and precipitation to account for the undulations in surface elevation of the finer-resolution ISM not resolved by the climate model[22]. In this scheme, surface air temperature ($T_a$) is shifted by $\Delta T_a = \gamma \Delta z$, with $\gamma = -0.008$ K m$^{-1}$ being the temperature lapse rate, and $\Delta z$ the difference in surface elevation between the ISM and the climate model. Precipitation is adjusted by a factor of $2^{\Delta T_a/10}$. Snowfall over the ice surface is calculated from monthly precipitation and surface air temperature using a temperature dependent ratio for the fraction of precipitation deposition as snow[22]. The ISM does not distinguish between snow, firn, and ice, assuming all snow deposited on the ice sheet's surface is immediately converted to ice. Ice surface melt is calculated from monthly surface air temperature using a positive-degree-day (PDD) scheme with a coefficient of 0.005 m per degree-day, but the reference temperature (TPDD) uses a reasonable offset instead of 0 °C, standing in implicitly for the net effect of omitted components of the surface energy balance such as radiative fluxes, as mentioned in last paragraph of this section. Liquid including meltwater and rainfall is assumed to immediately percolate downwards into the local ice column and exchanges its latent heat with the sensible heat of the next lowest layer. Any liquid that makes its way to the base is recorded as mass loss due to basal melt. Processes that may drive the marine ice sheet into a runaway slumping and collapsing—a mechanism coined the "Marine Ice Cliff Instability" (MICI)—are implemented via hydro-fracturing and ice cliff failure parameterization schemes constrained by modern and paleoclimate records[19,34]. The AIS and the GIS are modeled using the same ISM but with different spatial resolutions, 20 km for the AIS and 10 km for the GIS. With the hybrid ice dynamics with an imposed grounding line mass flux condition, our ISM simulates grounding line migration with satisfactory performance with these coarse resolutions[19].

Basal sliding is modeled for the AIS only, occurring where the ice sheet's basal temperature reaches the pressure-melting point. Sliding coefficients of the bed are obtained in an inverse ISM simulation driven by quasi-preindustrial climate (CERA20C[45] 1901–1920 climatology), in which the sliding coefficient at each grid point is adjusted iteratively until the local ice thickness equilibrates toward the present-day observed value[46]. When forced by present-day atmospheric fields (ERA5[47] 1981–2010 climatology), the ISM with no TPDD offset gives a total surface melt rate over all Antarctic ice shelves close to its modern estimation of 100 Gt/year[48]. TPDD is thus set to 0 °C for the AIS. The ISM uses a simple parameterization scheme for basal melt rates, which assumes a quadratic dependence on the 400 m ocean temperature above the pressure melting point of ice ($T_o - T_f$):

$$\mathrm{OM} = \mathrm{OMF}\left(\frac{K_T \rho_w C_w}{\rho_i L_f}\right)|T_o - T_f|(T_o - T_f) \qquad (1)$$

where $\rho_w$ is the density of sea water, $C_w$ is the specific heat capacity of sea water, $\rho_i$ is the density of ice, $L_f$ is the latent heat of fusion for ice, $T_o$ is the ocean temperature at 400 m, $T_f$ is the depth-dependent freezing point at the base of ice shelf, $K_T$ is a default coefficient for ocean-ice turbulent heat transfer. OMF is a spatially-independent coefficient, and is tuned so that under the present-day climate, the modeled basal melt rate of Antarctic ice shelves falls within the observational range. Under the present-day ocean temperatures (WOA2018 1981–2010

climatology) with an OMF of 4, the Antarctic ice shelf basal melt rate totals -1584 Gt year$^{-1}$ in the ISM, fitting in well the observational range of 1500 ± 237 Gt year$^{-1}$ [49].

As the only parameter being tuned for the GIS, TPDD is adjusted so that the ISM models a GIS volume close to present-day observation under quasi-preindustrial conditions (CERA20C 1901–1920 climatology). These experiments are carried out under the assumption that the present-day GIS has not changed substantially from its preindustrial state under the warming climate, so that much of its committed changes have not been fulfilled. An array of TPDD values are used for a series of preindustrial control simulations lasting 100,000 years, while TPDD = −4.0 °C results in a modeled near-equilibrium GIS volume close to modern observation (7.5 m versus 7.4 m SLE, Supplementary Fig. 5). This is likely due to the PDD scheme not accounting for the effect of insolation, and in order to reproduce the observed GIS a low TPDD value that compensates the omission of insolation is necessary. More details about the ISM and its tuning processes are available in publications describing the model[22] and using the model for future projection of the AIS[19,25].

## Climate model

The climate model coupled to PSUICE3D is the UVic Earth System Climate Model (UVic-ESCM)[23] version 2.8. UVic-ESCM consists of an energy-moisture balance atmospheric model, a three-dimensional ocean general circulation model, a thermodynamic/dynamic sea ice model, and a land model. Marine and land biogeochemical components of UVic-ESCM are turned off in this study as the focus here is on ice sheet-climate interactions. The model runs at a horizontal resolution of 3.6° × 1.8° (longitude-latitude) for its atmosphere, ocean, and sea ice components, with 19 vertical levels for the ocean model. Atmospheric heat and moisture transports are parameterized as diffusion processes, and precipitation occurs when relative humidity exceeds 85%. Temperatures over the land surface are calculated assuming a constant lapse rate. UVic-ESCM does not explicitly model ice sheets but assigns the surface type and surface elevation accordingly for grid cells occupied by an ice sheet. Radiative forcing due to changes in the atmospheric $CO_2$ level is parameterized through modification of the outgoing infrared flux, which also takes into account the water vapor feedback. The ocean component of UVic-ESCM, based on the Geophysical Fluid Dynamics Laboratory Modular Ocean Model 2.2 (GFDL-MOM2.2), uses prescribed present-day wind fields as a surface input, while an empirical relationship between atmospheric surface temperature and density helps introduce a dynamical wind feedback. The sea ice model uses elastic-viscous-plastic rheology to represent sea ice dynamics. Employing this reduced-complexity and computationally inexpensive climate model facilitates running a large number of multi-centennial experiments for various warming scenarios, while still showing good agreement with observations and having adequate capacity in modeling large-scale features associated with ice sheet freshwater-climate feedbacks. UVic-ESCM has been employed in a number of studies exploring long-term climate changes and paleoclimates[50,51], and for assessing the stability of AMOC and oceanic responses to freshwater forcings[11,52]. Without using flux adjustments like early coupled atmosphere-ocean climate models, UVic-ESCM nonetheless performs well in reproducing historical temperature changes and its modeled oceanic tracers display reasonable fidelity to observations[26].

## Emulating models with a different ECS using UVic-ESCM

Metrics quantifying the sensitivity of the GMSAT to changes in $pCO_2$ include the transient climate response (TCR)−defined as the change in GMSAT at the time $pCO_2$ doubles in the 1pctCO2 experiment, in which $pCO_2$ grows by 1% per year and doubles in 70 years. Another is the equilibrium climate sensitivity (ECS)−the eventual rise in GMSAT after doubling $pCO_2$ as the climate reaches a new equilibrium after

millennia, when slow components of the climate system such as the deep ocean have fully responded. While the ECS can be obtained by equilibrating climate models on millennia time scales[53], it can also be inferred in shorter transient simulations[54]. Estimates of the likely range of ECS have remained between 1.5 °C to 4.5 °C for decades since Jule Charney's 1979 report[55]. The sixth Assessment Report of the Intergovernmental Panel on Climate Change (IPCC-AR6) puts the likely range of ECS between 2.5 °C to 4.0 °C, and the very likely range between 2.0 °C and 5.0 °C[27]. In standard idealized abrupt-2xCO2 and abrupt-4xCO2 experiments, we restart UVic-ESCM from year 2000 of its 280 ppm-$CO_2$ control simulation but with doubled (560 ppm) and quadrupled (1120 ppm) $pCO_2$ respectively. UVic-ESCM displays an ECS of 3.4 ± 0.1 °C in abrupt-2xCO2 and abrupt-4xCO2 experiments (Supplementary Figs. 1 and 2). This is well within the likely range of 2.5–4 °C as assessed by IPCC-AR6[27] and is close to the best estimate value (3.0 °C) for the ECS. Due to strong nonlinearity of the AIS' responses to climate warming, even under the same future greenhouse gas emission scenario, differences between CMIP6 climate models give rise to a wide spread in projected future warming over Antarctica, resulting in substantial uncertainty in the rate of future Antarctic ice loss and its associated freshwater flux[25]. It is worth investigating the nature of ice sheet FWF-climate feedbacks in climate models with different ECS, but technological complexities including model structure, programing languages, platforms, etc. prohibit coupling with many different climate models. Here we take an alternative approach to explore ice sheet-climate interactions in scenarios with a different ECS. For a target ECS different from the UVic-ESCM's intrinsic ECS, rather than finding a climate model with such a trait, we still use the same UVic-ESCM, but emulate the suppositional climate model by scaling up or down the respective $pCO_2$ pathway. Denoting the target ECS as ECS*, and UVic-ESCM's native ECS as $ECS_0$ (3.4 °C), $pCO_2(t)$ the standard time-dependent $CO_2$ concentration (in ppm) for a specific scenario, then the scaled $CO_2$ concentration is:

$$pCO_2^*(t) = 280\left(\frac{pCO_2(t)}{280}\right)^{ECS^*/ECS_0} \quad (2)$$

$CO_2$ concentration can be scaled this way because the outgoing long-wave radiation (OLR) approximately changes linearly with surface temperature and the logarithm of atmospheric $CO_2$ concentration[56]. As a proof of concept, we conduct both abrupt-2xCO2 and abrupt-4xCO2 experiments with $CO_2$ concentrations scaled to represent a low ECS of 1.8 °C and a high ECS of 5.6 °C, respectively. By year 1500 of the abrupt-2xCO2 experiment, when the climate approaches a quasi-equilibrium, the rise in GMSAT (ΔGMSAT) is 1.85 °C, 3.41 °C, 5.63 °C respectively for ECS = 1.8 °C, ECS = 3.4 °C, and ECS = 5.6 °C. For the abrupt-4xCO2 experiment, ΔGMSAT is 3.63 °C, 6.77 °C, and 10.79 °C respectively. These experiments display good performance of this $CO_2$-scaling scheme in emulating climate models with alternative ECS.

IPCC-AR6 assesses that the likely range for the ECS is 2.5–4 °C. In this study, however, we scale the $CO_2$ pathways of future emission scenarios to emulate an ECS as high as 5.6 °C. The rationale for this approach is that UVic-ESCM displays a weaker polar amplification compared to more sophisticated coupled climate models. The amplification factor (AF) is defined as the change in SAT (ΔSAT) at each grid point with respect to its preindustrial condition divided by the changes in GMSAT (ΔGMSAT) when the climate reaches a quasi-equilibrium after doubling $pCO_2$:

$$AF = \frac{\Delta SAT}{\Delta GMSAT} \quad (3)$$

AF averaged over the Antarctic (south of 65°S) and the Arctic (north of 65°N) regions is 1.07 and 1.52 respectively for the UVic-ESCM, smaller than the multi-model-mean values of 1.49 and 2.16 calculated with 14

LongRunMIP[53] climate models (Supplementary Figs. 3 and 4). Long-RunMIP multi-model-mean polar AFs are stronger than those of UVic-ESCM by a factor of ~1.4. Due to the weak polar amplification in UVic-ESCM, we use an emulated ECS of 5.6 °C (i.e., 1.4 × 4.0 °C) in combination with the SSP5-8.5 pathway as a representative worst case scenario for future warming and ice sheet retreat.

## Ice sheet-climate coupling

Experiments of this study are carried out with a coupling time interval of 1 year, enabling a near-synchronous coupling between the ISM and the climate model (CM). Earlier studies of ice sheet-climate interactions were often conducted with an offline approach, whereby a prescribed change in the climate/ice sheet is used to drive an ice sheet/climate model, and changes in the ice sheet/climate model's output are fed back to a climate/ice sheet model to study the responses of the climate/ice sheet. This approach has strength in its simplicity and the identification of model responses is straightforward. Offline-coupled models indeed have revealed important feedback mechanisms and raised concerns over the net effect of ice sheet-climate feedbacks on the retreat of Earth's ice sheets[14,15,17,19]. However, in a typical offline-coupled ISM-CM workflow, each component is provided with relevant output from the other as prescribed fields throughout the simulation, even when the simulation covers a multi-centennial time span. Therefore, the strength of feedbacks cannot be fully assessed in experiments using offline-coupled models, as one component of the coupled ice sheet-climate system cannot respond to changes in the other in a temporally realistic way, which may underestimate/overestimate the strength of positive/negative feedback. Unlike previous studies that employed offline coupling or one-way forcing[15,17], our near-synchronous coupling between ISM and CM (with a coupling interval of 1 year) enables responses from one model to feedback to the other in time.

The ISM and the CM are fused by a `Python` coupler that reads necessary variables from one model's output and calculates boundary conditions needed by the other. Variables from the CM including near-surface air temperature, precipitation, and ocean temperatures at the 400 m depth are passed to the ISM for calculation of surface and basal mass balance as well as the ice sheet's internal thermal structure. The CM in turn needs ice coverage and surface elevation from the ISM as its surface boundary conditions. This coupling framework naturally resolves the decrease/increase in surface temperature associated with the increase/decrease in surface elevation as the ice sheet grows/decays. The coupled model also resolves the "elevation desert" effect and its associated feedback by adjustment of precipitation based on surface air temperature. Surface types perceived by the CM are calculated from modeled ice sheet extents, naturally resolving the ice-albedo feedback associated with changes in ice sheet geometry. In the current coupling scheme, the latent heat for surface and basal melting of ice sheets and ice shelves is not passed to the CM, although the latent heat for melting icebergs (treated as sea ice here) is passed to the CM. Mass (water) fluxes associated with the growth and decay of ice sheets are not passed to the CM, which has an ocean component model that has a rigid lid hence a constant volume.

In the UVic-ESCM, precipitation over land returns to the ocean instantaneously at discharge grid points of respective river drainage basins. In the coupled model, to avoid duplicated freshwater accounting, precipitation over Greenland and Antarctica is no longer routed to the ocean via the river model of UVic-ESCM, but is passed to the ISM and eventually discharged to the ocean as ice sheet freshwater flux. Ice sheet freshwater discharge is divided into liquid (meltwater and rainfall) and solid (icebergs) parts. The ISM calculates surface and basal melt rates of the ice sheet (and its ice shelves in the case of Antarctica), and routes meltwater to the ice sheet's edge grid cells. Calving rates at ice shelf fronts are parameterized based on a scheme accounting for hydrofracturing by meltwater produced on the ice

surface[19,22]. Due to the difference in horizontal resolution, the grid cells of UVic-ESCM and PSUICE3D are not collocated. Freshwater fluxes on the finer ISM grid are aggregated to the nearest UVic-ESCM coastal Greenland/Antarctica grid cells (Supplementary Fig. 6). Ocean grid cells and cells for ice sheet freshwater discharge are kept invariant in the coupled model, despite the fact that new ocean areas may appear/disappear as the marine ice sheet retreats/advances. The coupled ISM-CM does not account for changes in land-sea configurations due to sea level changes associated with thermal expansion of sea water, land ice melt, or perturbations to gravitational fields by redistribution of land ice mass[57]. Glacial isostatic adjustment is modeled in the ISM but is not passed to the CM due to the fixed land-sea configuration.

Ice sheet freshwater fluxes are imposed as salinity fluxes to the ocean model's top-level grid cells in UVic-ESCM. Iceberg drift and decay are not explicitly modeled here, and freshwater fluxes in the solid form are converted to sea ice growth in cells of discharge. This simple scheme conserves freshwater and energy budgets, but without transport of freshwater by longer-lived icebergs it may result in a general poleward shift of ice sheet freshwater input in the CM. As sea ice is thinner and more extensive than icebergs of the same volume, this approach may also introduce a cooling bias due to the ice-albedo feedback associated with the sea ice surface. Modeling iceberg drift and decay, however, is beyond the scope of this study as it requires spatial and temporal resolutions higher than those provided by the UVic-ESCM.

CERA20C[45] 1901–1920 climatological monthly surface air temperature and precipitation and WOA2018[58] 1981–2010 climatological annual ocean temperature are combined as an "observational baseline climate", which is used to drive the ISM's tuning and control simulations. UVic-ESCM also runs a preindustrial control simulation with 280 ppm atmospheric $p\mathrm{CO}_2$ to get a "model baseline climate". In the coupled ISM-CM, climate anomalies relative to the model baseline are added to the observational baseline[25]. Temperature anomalies are added directly to the model baseline, while precipitation anomalies ($\Delta P$) are added as a fractional change from the baseline precipitation $P_0$ parameterized via the change in surface air temperature $\Delta T_a$: $\Delta P = P_0 2^{\Delta T_a/10}$. The proportion of precipitation deposited to ice surface as snow is still computed using the ISM's surface mass balance scheme[59]. This is essentially a bias-correction method for the UVic-ESCM, in that only changes relative to its modeled preindustrial climate take effect for coupled ISM-CM simulations, while any bias in UVic-ESCM's simulated climatological fields are removed.

The coupled ISM-CM has a reasonably good performance in simulating the present-day states of both ice sheets and trends in ice sheets' volume over recent decades (Supplementary Figs. 18–22). Notable biases are present for the AIS, e.g., the coupled ISM-CM simulates too extensive ice shelves and too little surface melt under modern climate conditions. The modeled Antarctic ice loss is generally slower than that estimated by IMBIE 2021. In addition to UVic-ESCM's own limitations, decadal and multi-decadal climate variability over recent decades, which cannot be reproduced by the coupled ISM-CM, may be partially responsible for the differences.

## Design of numerical experiments

A series of experiments, illustrated in the flow chart of Supplementary Fig. 7, are carried out with standalone UVic-ESCM, standalone PSUICE3D for Greenland and Antarctica, and the coupled ISM-CM. In all experiments, Earth's orbital parameters are kept constant at the present-day values. Atmospheric $CO_2$ concentration is 280 ppm for UVic-ESCM and coupled model preindustrial simulations, and varies in the *historical* and six shared socio-economic pathways (SSPs) as specified by CMIP6[60]. Radiative forcings from other greenhouse gases are not modeled by UVic-ESCM, so the warming in future scenarios simulated by this model is expected to be slightly lower than a CMIP6 model with the same ECS. The ice-albedo feedback, the temperature-

elevation feedback, and the "elevation desert" effect are naturally resolved in the coupled model. They are not turned off in experiments although the focus of this study is on the feedbacks between ice sheet FWF and the warming climate.

Numerical experiments in this study (Supplementary Table 1) can be categorized as among four types:

(1) (Exp. 0–5) Standalone ISM/CM tuning, sensitivity test and control simulations. These are designed for getting the model baseline climate and initial conditions for the coupled ISM-CM simulations.

(2) (Exp. 6–9) Coupled ISM-CM control simulations with the atmospheric $CO_2$ level fixed at 280 ppm. These experiments are designed for setting up initial conditions for historical-future simulations.

(3) (Exp. 10–15) Coupled ISM-CM single-member simulations with transient $CO_2$ levels specified in one of the six anthropogenic warming scenarios. These experiments, in which FWF from both ice sheets are interactive, are designed for investigating the dependence of ice sheet FWF-climate feedbacks on warming scenarios. The CMIP6 *historical* scenario (1850–2014) is combined with six socio-economic pathways (2015–2500) to form six historical-future scenarios covering 1850–2500. Each suite of experiments is carried out with three emulated ECS (3.0/4.0/ 5.6 °C) to further diversify future warming scenarios.

(4) (Exp. 16–23) Coupled ISM-CM 10-member ensemble simulations in a moderate warming scenario (historical-SSP2-4.5 with ECS = 4.0 °C) and an intensive warming scenario (historical-SSP5-8.5 with ECS = 5.6 °C). Each suite of experiments is carried out with four configurations for ice sheet FWF. These experiments are designed for investigating potential dependence of ice sheet-climate feedbacks on the pace of warming.

Four configurations of the coupled ISM-CM are used to distinguish the effect of FWF from either/both ice sheet. A single run in each model configuration is sufficient for identifying the differences between a pair of configurations when the changes in ice sheets and the climate are large, but the differences between configurations during the early stage of warming are small and are strongly influenced by natural variability. To obtain reliable signals in moderate warming scenarios or in early stages of intensive warming scenarios, it is necessary to conduct ensemble simulations to suppress noise from internal variability. For each model configuration and each warming scenario, 10 ensemble members are initiated from different time slices (each separated by 50 years) of the preindustrial control simulation over a time span of 500 years (year 3500–4000). This allows for a subdued interference from internal oscillations in the ensemble-mean, as a result of largely canceling phases between these simulations.

### Feedback analysis

The sign and strength of ice sheet FWF-climate feedbacks can be quantified using a feedback factor. The feedback factor $\gamma$ is defined following ref. 61's recommendation for non-radiative processes:

$$\gamma = \frac{\dot{V}_{iFWF} - \dot{V}_{cFWF}}{\dot{V}_{iFWF}} = 1 - \frac{\dot{V}_{cFWF}}{\dot{V}_{iFWF}} \qquad (4)$$

where $\dot{V}_{iFWF}$ is the rate of change in the AIS' volume in coupled ISM-CM simulations with interactive ice sheet FWF (the total response), $\dot{V}_{cFWF}$ is the rate of change in the AIS' volume in coupled simulations with constant ice sheet FWF at pre-industrial conditions (the reference response). The perturbation is defined as the change in GMSAT relative to pre-industrial (ΔGMSAT). $\dot{V}_{iFWF}$ and $\dot{V}_{cFWF}$ are one-to-one matched based on ΔGMSAT, so the reference response can be interpreted as the rate of ice loss with the same perturbation in GMSAT but no interactive ice sheet FWF. In practice, $\dot{V}_{iFWF}$ and $\dot{V}_{cFWF}$ are sampled and matched

from bins of GMSAT with a width of 0.25 K. The centers of GMSAT bins range from 288 K to 300 K on an interval of 0.25 K.

## Data availability

All data needed to evaluate the conclusions in the paper are present in the paper and/or the Supplementary Information. Animations of transient changes in selected key variables for moderate/intensive warming scenarios are available in Supplementary Videos. Output from the coupled ice sheet-climate model has been deposited in the Zenodo repositories https://doi.org/10.5281/zenodo.10988828 (https://zenodo.org/records/10988828) and https://doi.org/10.5281/ zenodo.10988830 (https://zenodo.org/records/10988830).

## Code availability

Ice sheet model codes are available from the authors. UVic-ESCM is available freely from the University of Victoria (http://terra.seos.uvic. ca/model), and information about modifications to UVic-ESCM for coupling with the ice sheet model is available from the corresponding author. Jupyter-notebook scripts for processing the coupled model output and making figures of the main text are available from the Zenodo repository https://doi.org/10.5281/zenodo.10988828 (https:// zenodo.org/records/10988828).

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

## Acknowledgements

The authors thank University of Victoria for developing the UVic-ESCM model, and Maria Rugenstein and Jonah Bloch-Johnson for organizing the LongRunMIP project. D.L. thanks Michael L. Bender and Meng Zhou for helpful discussions on this research. The computations in this work were carried out on the cluster supported by the Center for High Performance Computing at Shanghai Jiao Tong University. The study was supported by the National Natural Science Foundation of China (42488201 Y.H. and D.L.), the National Key Research and Development Program of China (2023YFF0805201 D.L.), the Natural Science Foundation of Shanghai (22ZR1430500 D.L.), the Oceanic Interdisciplinary Program of Shanghai Jiao Tong University (SL2023MS015 D.L.), and the US National Science Foundation (2035080 and 1934477, R.M.D. and D.P.).

## Author contributions

D.L. conceived the model experiments and the ice sheet-climate coupling framework with conceptual input from R.M.D., D.P., and Y.H.; R.M.D. and D.P. developed the ice sheet model codes; D.L. performed model experiments, analyzed model output, produced figures, and drafted the manuscript; D.L., R.M.D., D.P., and Y.H. participated in the data interpretation and contributed to editing the manuscript.

## Competing interests

The authors declare no competing interests.
