## [Peer Review File · Nature Communications]

Competing climate feedbacks of ice sheet freshwater discharge in a warming worldREVIEWER COMMENTS

Reviewer #1 (Remarks to the Author):

The authors utilize coupled climate ice sheet model simulations to reveal feedbacks between Greenland and Antarctic ice sheet melt and the coupled ocean-atmosphere-ice system. They find that Greenland melt has a cooling effect on the ocean, which in turn suppresses Greenland melt through a negative feedback. Antarctic melt tends to warm the ocean adjacent to the ice shelves, further enhancing melt in a positive feedback. However, in the case of extreme Antarctic melting, the positive feedback in Antarctica becomes negative. The authors also show that enhanced melting from either ice sheet acts to suppress melting of the other ice sheet.

The results are noteworthy in that they reveal feedbacks associated with ice sheet melt that can only be captured in coupled climate systems. The authors show how meltwater climate feedbacks can be of opposite sign for the Greenland and Antarctic ice sheets, and they reconcile previous studies that showed either positive or negative feedbacks associated with Antarctic melting. Additionally they show how melt from each ice sheet can feed back on the climate-ocean-atmosphere system and act to suppress mass loss from the other ice sheet. The work is a significant contribution to our understanding of ice sheet-climate feedbacks and their role in climate change and sea level rise.

The authors have carefully considered various feedbacks and sources of uncertainty in support of their conclusions, and the data analysis, interpretation and conclusions are solid.

I have some specific suggestions on the manuscript attached, but I feel that the manuscript is ready to be published with minor revisions.

Reviewer #2 (Remarks to the Author):

The future melt of the Greenland and Antarctic ice sheets is the main uncertainty for projection of sea level rise. One of the critical points is to account for the feedbacks between climate and ice sheets first, as it implies complex processes on a wide range of scales and second, as most climate models (including those in CMIP6) do not have an interactive ice sheet component and are thus not able to represent explicitly those interactions. The authors propose to reduce those uncertainties by performing and analyzing probably what is the largest set of experiments of future changes with a climate model synchronously coupled to an ice sheet model. The experimental design is clear, the analyses and the interpretation of the results relatively straightforward and the text easy to follow. One of the most interesting results is that the magnitude of the ice-sheet climate feedbacks (and even the sign of the sum of the feedbacks) depends of the forcing and of the climate sensitivity of the model. This study is also one additional reason to strongly push to continue the development of climate-ice sheet models and to use them more systematically in projections. Overall, I consider thus that this study is an interesting contribution to the topic. However, the authors should be clearer on the limitations of the model they use and I would also suggest to be more quantitative in the evaluation of the feedbacks as detailed below.

1/ The authors use an Earth system Model of Intermediate Complexity (EMIC), which has a coarser resolution and simplified physics, in particular in the atmosphere, compared to the state-of-the-art General Circulation models (GCM). I have no problem with that as it is currently impossible to perform the number of simulations needed in this study with a GCM. However, the authors must be clearer on this and more explicit on the limitations and uncertainties this brings. First, they should state explicitly when the model is introduced as 'climate model' (line 141 -last paragraph of the introduction) that it is an EMIC since 'climate model' is vague and could be interpreted differently. The uncertainties should also be made more explicit as the ice sheet climate interactions depend on processes that could not be resolved at the coarse resolution of the ocean and by the (very simplified) atmosphere model applied

here. Consequently, the response to freshwater input can be strongly model dependent, high resolution model having in some regions a response to freshwater input of a different sign compared to a low resolution version and thus potentially a different sign for some feedbacks there (e.g., Beadling, et al. 2022). Specifically, the melting in ice sheet cavities is influenced by exchanges at the shelf break, by the circulation within the cavity, by the changes in the salinity on the continental shelf... This is represented by a simple quadratic formulation in the model used here. This is probably the best choice at this resolution but the associated limitations should be explained when the representation of ice shelf melting is discussed. If I have understood correctly, meltwater from ice sheet and ice shelf melting is injected at ocean surface but in the real world a large fraction is injected at depth. This could have a strong impact on the oceanic stratification, on the warming at depth and thus on the positive feedback described in the paper (see for instance Mathiot et al. 2017; Jourdain et al. 2017; Mackie et al. 2020). The final paragraph of the manuscript describes somehow those limitations but without details on the processes that may have been missed or key references on the subject, except for the representation of icebergs. This should be expanded in the revised version so that the reader can have a better knowledge of the limitations of this study and of what can be brought in the future when new tools with a better representation of those processes will be available. I would also change the last sentence 'are likely to be confirmed in future work' because it is impossible to know at this stage how those processes poorly represented in this model affect the conclusions.

2/ The paper deals with ice sheet climate feedbacks but those feedbacks are never really quantified. The difference between experiments is discussed. The sign of the feedback is addressed but there is no precise quantification of the feedback strength. From the experimental design (or with a few additions), it should be possible to quantify the feedbacks using a classical measure such as the feedback factors or the feedback gain, which are widely used for radiative feedbacks but can also be computed for other feedbacks (e.g., Roe 2009, Goosse et al. 2018). This would also provide numbers that can be compared with the results of future studies to determine qualitative differences (i.e. different sign of the feedback) but also quantitatively.

Specific points

Line 218. It is not clear for me from the main text nor from the appendix how the 'fixed ice sheet FWF' are computed and how they compare to the ones computed interactively (Fig 1ab). Is it the FWF fluxes from the climate model only? Are they derived from an estimate for present-day conditions? The tar files I uploaded includes movies but I have not seen any reference or explanation for those movies.

References

- Beadling, et al. 2022. Importance of the Antarctic Slope Current in the Southern Ocean response to ice sheet melt and wind stress change. *Journal of Geophysical Research: Oceans*, 127, e2021JC017608. <https://doi.org/10.1029/2021JC017608>
- Goosse H et al., 2018. Quantifying climate feedbacks in polar regions. *Nature Communications* 9, 1919, DOI: 10.1038/s41467-018-04173-0
- Jourdain, et al. 2017. Ocean circulation and sea-ice thinning induced by melting ice shelves in the Amundsen Sea, *J. Geophys. Res. Oceans*, 122, 2550–2573, doi:10.1002/2016JC012509.
- Mackie et al. 2020. Climate Response to Increasing Antarctic Iceberg and Ice Shelf Melt. *Journal of Climate* 33, 8917 DOI: 10.1175/JCLI-D-19-0881.1
- Mathiot et al. 2017. Explicit representation and parametrised impacts of under ice shelf seas in the zcoordinate ocean model NEMO 3.6. *Geosci. Model Dev.*, 10 (7) (2017), p. 2849
- Roe, G. H., 2009 Feedbacks, timescales and seeing red. *Annu. Rev. Earth. Planet. Sci.* 37, 93–115. <https://doi.org/10.1146/annurev.earth.061008.134734>

Reviewer #3 (Remarks to the Author):

Review of

Competing climate feedbacks of ice sheet freshwater discharge in a warming world

by Sawei Li and others

General

This study uses an asynchronously (once-per-year) coupled climate-ice sheet model to quantify the feedbacks of freshwater release from the ice sheets on future ice sheet mass loss. The processes that are most relevant and under investigation here are enhanced stratification of the upper ocean, cooling the upper ocean and atmosphere (negative feedback), and warming the deeper ocean layers (positive feedback). Then there is the weakening of the AMOC, reducing interhemispheric heat exchange by the ocean (bipolar see-saw). The technical quality of the paper is good: it is well-written with high-quality figures, and the results and acting processes are clearly explained. My main concern is that the adopted highly idealized/simplified parametrizations in the low-resolution climate model could lead to very large uncertainties in the modeled ice sheet mass balance, to the point that I question whether the approach is sufficiently robust to answer the main research questions. The authors are aware of these potential limitations and rightfully point out that it will take many years before full-fledged, high-resolution ESMs can be run for such extended periods.

Major comments

I acknowledge the need for fast climate models for these applications involving ice sheet-ocean interactions, and I appreciate the efforts of the authors to (asynchronously) couple climate and ice sheet models. However, my main concern after reading this manuscript is that the climate model used is so coarse (order several 100s km horizontal resolution), and its forcings and parametrizations to calculate moisture and heat transport in ocean, atmosphere, ice sheets, and the interfaces between them so simplified (e.g., basal and surface melt) that it is unclear whether it is at all possible to robustly model ice sheet climate interactions (see description of the Climate Model line 898 and further). Will this model setup really resolve the "contradicting conclusions" (l. 120) of previous work done? The authors claim that (l. 494): "... first order features identified from this study, including the inter-hemispheric interaction between retreating ice sheets, the competing positive and negative feedbacks, the dependence of these feedbacks on the pace of warming, and positive feedback diminishing as ice shelves are lost, are likely to be confirmed in future work." But there is no way to really tell. A concrete, bare minimum improvement of the manuscript should include a (spatially distributed, not spatially integrated) model evaluation to demonstrate that the contemporary mass balance of the ice sheets is well captured, as well as their first-order sensitivity to atmospheric (Greenland) and oceanic (Antarctica) warming, as observed in the last decades. This includes a separate evaluation of the main mass balance components, i.e., surface mass balance (snowfall, runoff), ice discharge (ice velocity, ice thickness, basal melt of Antarctic ice shelves), and their recent trends that invoked the contemporary mass loss of both ice sheets.

In the introduction, frequent reference is made to glacial-interglacial cycles and the associated meltwater pulses. But today's climate and ice volume are very different from those during glacial epochs (higher temperatures, smaller ice volumes), and the expected response to meltwater pulses is therefore (very) different. Please comment on whether/why these comparisons are still warranted.

Fig. 1: Can you explain is the AIS response so much less regular than the GrIS response? From the paleo-record, we know that the GrIS climate is very sensitive to changes in the AMOC strength.

Minor and textual comments

The first line of the abstract: the list of processes that influence freshwater fluxes from ice sheets is

incomplete and appears rather arbitrary. I suggest only mentioning the feedbacks central to this paper.

I. 113: "Surface cooling reduces ice surface melting and ice shelf crevassing and calving..." The impact of atmospheric cooling on ice shelf crevassing seems limited to me. Are the authors referring here to the hydrofracturing of ice shelves following meltwater ponding?

I. 131: "... in the existing offline coupling framework..." Currently, various fully coupled models are being tested and/or run (e.g., UKESM). Please mention these.

NCOMMS-23-27716-T

Competing climate feedbacks of ice sheet freshwater discharge in a warming world

RESPONSE TO REVIEWS

We much appreciate the time and effort the reviewers have dedicated to provide valuable feedback on the manuscript. We are grateful to the reviewers for their timely, insightful, and constructive comments. I would like to apologize for the delayed response. The manuscript has been revised accordingly, with major changes to the manuscript and Extended Data listed below:

- A new figure (Fig. 4) is included in the main text for feedback analysis, Methods and the main text have been updated as well for describing the feedback analysis and discussing the findings.
- The Discussion section has been substantially expanded to discuss limitations of the models and potential impacts of the processes that cannot be well-represented in the current modeling toolkit.
- Five new figures are included in Extended Data for model evaluation, so that readers could have a how the models perform in reproducing observed states and trends of ice sheets, and the study's limitations.
- Numerous textual edits following reviewers' suggestions to improve clarity.

Please find the point-by-point responses in the following pages, where the reviewers' comments are in *blue italic*, and our responses in black text. A PDF manuscript file with changes marked by `latexdiff` is included with the re-submission.

Dawei Li, On behalf of all co-authors,

April 17, 2024

Reviewer #1

The authors utilize coupled climate ice sheet model simulations to reveal feedbacks between Greenland and Antarctic ice sheet melt and the coupled ocean-atmosphere-ice system. They find that Greenland melt has a cooling effect on the ocean, which in turn suppresses Greenland melt through a negative feedback. Antarctic melt tends to warm the ocean adjacent to the ice shelves, further enhancing melt in a positive feedback. However, in the case of extreme Antarctic melting, the positive feedback in Antarctica becomes negative. The authors also show that enhanced melting from either ice sheet acts to suppress melting of the other ice sheet.

The results are noteworthy in that they reveal feedbacks associated with ice sheet melt that can only be captured in coupled climate systems. The authors show how meltwater climate feedbacks can be of opposite sign for the Greenland and Antarctic ice sheets, and they reconcile previous studies that showed either positive or negative feedbacks associated with Antarctic melting. Additionally they show how melt from each ice sheet can feed back on the climate-ocean-atmosphere system and act to suppress mass loss from the other ice sheet. The work is a significant contribution to our understanding of ice sheet-climate feedbacks and their role in climate change and sea level rise.

The authors have carefully considered various feedbacks and sources of uncertainty in support of their conclusions, and the data analysis, interpretation and conclusions are solid.

I have some specific suggestions on the manuscript attached, but I feel that the manuscript is ready to be published with minor revisions.

We sincerely thank the reviewer for providing thoughtful, constructive, and very detailed comments, which have been very helpful for improving the clarity of our manuscript.

1. Lines 93-94: Perhaps provide a brief description of what the Meridional Overturning Circulation does.

The following sentence has been added before discussing how ice sheet FWF may change the Meridional Overturning Circulation: *“In present-day climate conditions, warm, salty surface water flows northward in the Atlantic basin and are cooled in the high latitudes of the North Atlantic, forming deep waters that sink to depths and spread to the Southern Ocean, where they mix into the World Ocean. This Atlantic Meridional Overturning Circulation (AMOC) transports heat across the equator to the North Atlantic and influences inter-hemispheric energy balance of the climate system.”*

2. Lines 139-141: Although this is described in the methods section, it would be helpful to have

some additional details here, e.g. at what intervals the coupling happens, and the processes that are included in the ice sheet and climate models.

As suggested, the first two sentences of the paragraph have been expanded to: *“Here we provide numerical simulations with a three-dimensional ice sheet model (ISM) quasi-synchronously coupled to a climate model of reduced complexity with a coupling time step of 1 year. The ISM is a 3-D dynamic-thermodynamic model that simulates ice sheets’ surface and basal mass balance using bias-corrected climate fields from the climate model, as well as processes including basal sliding and bedrock deformation. The climate model is an Earth system model of intermediate complexity (EMIC) that includes a three-dimensional ocean model, a land model, and a two-dimensional energy-moisture balance model for the atmosphere. Using a reduced-complexity climate model enables carrying out a large set of multi-century scale simulations with different combinations of model configurations, climate sensitivities, emission scenarios, and initial conditions. Limitations of such model choices will be discussed in the Discussion section. The climate model’s output drives ice sheets’ mass balance and changes, while the ISM feeds back its simulated ice surface elevation and ice sheet FWF to the climate model. We focus on the interactions between ice sheet FWF and the climate for both the AIS and the Greenland Ice Sheet (GIS) in historical-future climate scenarios specified by six Shared Socioeconomic Pathways (SSPs).”*

3. Lines 152-154: I believe that in configuration #1, the ice sheet responds to climate change through mass loss over time, but the FWF passed to the ocean is the pre- industrial FWF. If so, please clarify this here.

The sentence has been changed to *“Ice sheet FWF passed to the ocean is kept constant at the preindustrial level for both ice sheets, though ice sheets respond to changes in the climate.”* We also add a sentence to clarify the configuration regarding to ice sheet surface elevation: *“In all configurations ice sheets can interact with the climate by feeding back their surface elevation to the climate model as a surface boundary condition.”*

4. Line 155/156: revise to read “FWF is fully interactive from both ice sheets”.

Done.

5. Line 157/158: Change “suppress noises” to “suppress noise”.

Done.

6. Line 178-184: The large spread doesn’t necessarily indicate that the polar climates are substantially different, though this is the most logical factor. I suggest clarifying this statement a bit.

The sentence has been changed to: *“Under the same emission scenario, the large spread in ECS leads to very different changes in GMSAT across CMIP6 climate models. Inter-model differences in projected polar temperatures are even greater due to “polar amplification”, which would result in divergent future trajectories of polar ice sheets when climate model outputs are used to drive ice*

sheet models.”

7. *Line 201/202: Clarify that this is the simulation with interactive FWF from both ice sheets.*

The sentence has been changed to “... the coupled ice sheet-climate model (with interactive FWF from both ice sheets) shows that ...”

8. *Line 215: Insert “the” before “mid-21st”*

Done.

9. *Line 218/219: Clarify that this is fixed preindustrial FWF.*

We have added the following sentence before this line to clarify what “fixed ice sheet FWF” means: “A group of simulations are carried out with “fixed ice sheet FWF”, i.e. ice sheet FWF received by the climate model is kept same as the long-term mean FWF simulated by the coupled ISM-climate model in pre-industrial conditions.”

10. *Line 225: Insert “the” before “AMOC”.*

Done.

11. *Figure 1: I suggest adding a legend for dashed, dashed-dotted, and solid lines on the figure itself rather than at the end of the caption, as this information was difficult to find.*

As suggested, we have added a legend on Fig. 1 for the line styles.

Also, if the best estimate for the IPCC-AR6 is an ECS of 3.0°C, I think that the thick solid lines that stand out more prominently should be used for this case.

The “best estimate” of the ECS for the IPCC-AR6 is indeed 3.0°C. UVic-ESCM, however, displays a ~ 30% weaker “polar amplification” compared with more sophisticated climate models (see Extended Data Fig. 4). Therefore, UVic-ESCM with an emulated ECS of 4.0°C simulates roughly the same degree of polar warming as typical CMIP6 models with an ECS of 3.0°C would do. This is why we use thicker lines for the case with an emulated ECS of 4.0°C.

Relevant discussion has been updated to clarify the reasoning for using 3.0°C, 4.0°C, and 5.6°C as three representative ECS: “Given the same increase in $p\text{CO}_2$, UVic-ESCM with an emulated ECS of 4.0°C would simulate polar warmings of roughly the same magnitude as typical CMIP6 models with an ECS around 3.0°C would do. Considering these factors, we scale the CO_2 levels in historical-future scenarios to emulate three representative ECS (3.0°C, 4.0°C, and 5.6°C), which produce roughly the same changes in polar temperatures as CMIP6 models with ECS of 2°C, 3°C, and 4°C, respectively (Methods).”

12. *Lines 231-238: As Greenland is sensitive to the atmosphere, I’m wondering about the pathway reducing GIS melt – is it through direct ocean influence or due to both the cooling atmosphere and*

cooling ocean?

It is likely to be an indirect pathway that GIS melt weakens the AMOC (Fig. 2i), reducing northward ocean heat transport to the North Atlantic and causing atmospheric cooling there.

13. Lines 252-255: The title seems misleading here, because the authors are discussing both the Greenland and Antarctic effects. I suggest revising to also include Greenland in the title.

Thanks for the suggestion, but please note that we have not included “Greenland” here because the competition between positive and negative feedbacks are only relevant for Antarctica in present and future climate scenarios. The presence of extensive ice shelves is necessary for the “ice shelf basal melt → ocean stratification → subsurface warming” positive feedback to work. The current Greenland Ice Sheet does not have an extensive ice shelf, so the effect of the positive feedback is minimal.

14. Line 270/271: add “of” before “0.1°C”.

Done.

15. Line 274: Which simulations are being referred to here – the simulations that include the AIS FWF?

Here the simulations refer to those with the Greenland FWF. To clarify, we have changed the sentence to: “*The Greenland FWF-induced surface cooling of ~ 1 °C is concentrated in the North Atlantic to the south of Greenland accompanied by weaker warming elsewhere (Fig. 3), lessening its global-mean impact.*”

Also change “are concentrated” to “is concentrated”.

Done.

16. Lines 285-287: Figure 2j can be referred to here.

Done.

17. Lines 287-292: Here figure 2k can be referred to.

Done.

18. Lines 305-307: Can the authors provide some evidence regarding this, perhaps just numbers to quantify the surface and subsurface effects?

We have added the following sentence here to be more quantitative: “*In year 2300 of simulations following the SSP5-8.5 scenario with an emulated ECS of 5.6°C, when the WAIS is under a runaway retreat, the surface cooling effect of ice sheet FWF exceeds 3°C over West Antarctica, while the subsurface warming approaches 3°C in the Weddell Sea (Fig. 3d).*”

19. Line 333: Change “sign-reverse” to “sign-reversal”

Done.

20. Line 359: Reference Figure 2i here.

Done.

21. Lines 362-364: Any thought as to the impact sea level changes might have if they were included?

We speculate that if sea level changes were included, retreat of the GIS raises sea level around Antarctica, which might accelerate the retreat of the marine portions of the AIS. The other way round, retreat of the AIS also raises sea level around Greenland, which might have minimal influences on the GIS except for some of its tidewater glaciers. Relative sea level fall due to GIA and lessening of the ice sheet’s gravitational attraction might have a stabilizing effect for an ice sheet under retreat.

22. Lines 418/419: Suggest changing to read “In S20 and D21, the studies that show a net negative feedback” to improve readability.

Done.

23. Lines 451/452: I believe the “faster ice sheet retreat” discussed is referring to the Antarctic ice sheet. Please clarify.

The phrase has been changed to “an overall faster retreat of the AIS and associated sea level rise” to improve clarity.

24. Lines 472/473: The authors don’t discuss the impact of the vertical profile of icebergs, which is not accounted for here, in driving freshwater flux. Perhaps the authors can comment on how this might affect the simulations.

Discussion on the current model’s limitations has been expanded and now includes an item for the effect of freshwater deposition at depth:

“• Depth of freshwater injection: *The coupling scheme of this study injects ice sheet FWF to the top layer of the ocean model, while in the real world basal meltwater is injected at depth by both ice shelves and icebergs. This modeling choice is due to the lack of explicitly modeled ice shelf cavities and icebergs. Strengthening of ice shelf cavity circulations due to intensified basal melting may deepen the mixed layer around Antarctica, in contrast to iceberg meltwater, which is shown to enhance stratification. Adding iceberg meltwater at depth was found to increase the magnitude of subsurface warming and sea ice trends. This complexity could affect the positive feedback identified in this and previous studies.*”

25. Line 791/792: Add “the” before “shallow ice approximation” and “shelfy-stream approxima-

tion”.

Done.

26. *Lines 829-833: The details of the PDD scheme are unclear. Could the authors provide a few additional details about the scheme?*

Description of the PDD scheme and surface mass balance calculations has been expanded:

“The ISM does not distinguish between snow, firn, and ice, assuming all snow deposited on the ice sheet’s surface is immediately converted to ice. Ice surface melt is calculated from monthly surface air temperature using a positive-degree-day (PDD) scheme with a coefficient of 0.005 m per degree-day, but the reference temperature (TPDD) uses a reasonable offset instead of 0° C, standing in implicitly for the net effect of omitted components of the surface energy balance such as radiative fluxes, as mentioned in last paragraph of this section. Liquid including meltwater and rainfall is assumed to immediately percolate downwards into the local ice column and exchanges its latent heat with the sensible heat of the next lowest layer. Any liquid that makes its way to the base is recorded as mass loss due to basal melt.”

27. *Line 847/848: Change “out ISM” to “our ISM”.*

Done.

28. *Line 867/868: Can the authors explain how the melt rate factor is applied? Is it a term in the quadratic equation described earlier?*

The melt rate factor is now described in more details: *“The ISM uses a simple parameterization scheme for basal melt rates, which assumes a quadratic dependence on the 400 m ocean temperature above the pressure melting point of ice ($T_o - T_f$):*

$$OM = OMF \left(\frac{K_T \rho_w C_w}{\rho_i L_f} \right) |T_o - T_f| (T_o - T_f)$$

where ρ_w is the density of sea water, C_w is the specific heat capacity of sea water, ρ_w is the density of ice, L_f is the latent heat of fusion for ice, T_o is the ocean temperature at 400 m, T_f is the depth-dependent freezing point at the base of ice shelf, K_T is a default coefficient for ocean-ice turbulent heat transfer. OMF is a spatially-independent coefficient, and is tuned so that under the present-day climate, the modeled basal melt rate of Antarctic ice shelves falls within the observational range.”

29. *Line 881/882: Change “changes has” to “changes have”.*

Done.

30. *Line 915: Change “over land surface” to “over the land surface”.*

Done.

31. Lines 944-946: Change “show a good performance” to “performs well”.

Done.

32. Lines 952-953: Please clarify what is meant by “these responses”.

The sentence has been changed to: “Metrics quantifying the sensitivity of the GMSAT to changes in pCO_2 include ...”

33. Line 954/955: I don't think the 1pcCO2 experiment has been explained so far, please explain briefly.

These lines have been rephrased to “... the 1pctCO2 experiment, in which pCO_2 grows by 1% per year and doubles in 70 years. Another is the equilibrium climate sensitivity (ECS) ...”

34. Line 964/965: Change “ECS stayed” to “ECS have remained”.

Done.

35. Lines 1061-1085: The difference between an offline-coupled model and the model used here are not entirely clear. I suggest clarifying the timescales and number of coupling steps in a typical offline-coupled simulation.

We have clarified relevant sentences as follows: “However, In an typical offline-coupled ISM-CM workflow, each component is provided with relevant output from the other as prescribed fields throughout the simulation, even when the simulation covers a multi-centennial time span. Therefore, the strength of feedbacks cannot be fully assessed in experiments using offline-coupled models, as one component of the coupled ice sheet-climate system cannot respond to changes in the other in a temporally realistic way, which may underestimate/overestimate the strength of positive/negative feedback. Unlike previous studies that employed offline coupling or one-way forcing, our near-synchronous coupling between ISM and CM (with a coupling interval of 1 year) enables responses from one model to feedback to the other in time.”

36. Lines 1098-1101: Do fluxes of mass and energy also pass from the ISM to the CM? Please clarify.

To clarify how energy fluxes are passed between the models, we have added the following sentence to the end of the paragraph: “In the current coupling scheme, the latent heat for surface and basal melting of ice sheets and ice shelves is not passed to the CM, although the latent heat for melting icebergs (treated as sea ice here) is passed to the CM. Mass (water) fluxes associated with the growth and decay of ice sheets are not passed to the CM, which has an ocean component model that has a rigid lid hence a constant volume.”

37. Line 1126/1127: Change “on ice surface” to “on the ice surface”.

Done.

38. Line 1133/1134: Change “despite” to “despite the fact that”.

Done.

39. Lines 1154-1158: As mentioned earlier could the authors comment on the impact of melting along deeper portions of the iceberg, which is not accounted for here?

In the revised manuscript, relevant discussion on the potential impact of iceberg freshwater deposition at depth is given in the main text Discussion section.

40. Lines 1172-1174: Suggest revising to read “Temperature anomalies are added directly to the model baseline, while precipitation anomalies (DP) are added as a fractional change...”

Done.

41. Line 1234 and 1241: Change “suite of experiments are” to “suite of experiments is”

Done.

42. Lines 1243-1247: This statement is somewhat unclear. Can the authors clarify?

The sentence has been changed to: “These experiments are designed for investigating potential dependence of ice sheet-climate feedbacks on the pace of warming.”

43. Line 1258/1258: Change “ “noises” ” to “noise”, without quotes.

Done.

44. Line 1260: Change “internal variabilities” to “internal variability”.

Done.

Reviewer #2

The future melt of the Greenland and Antarctic ice sheets is the main uncertainty for projection of sea level rise. One of the critical points is to account for the feedbacks between climate and ice sheets first, as it implies complex processes on a wide range of scales and second, as most climate models (including those in CMIP6) do not have an interactive ice sheet component and are thus not able to represent explicitly those interactions. The authors propose to reduce those uncertainties by performing and analyzing probably what is the largest set of experiments of future changes with a climate model synchronously coupled to an ice sheet model. The experimental design is clear, the analyses and the interpretation of the results relatively straightforward and the text easy to follow. One of the most interesting results is that the magnitude of the ice-sheet climate feedbacks (and even the sign of the sum of the feedbacks) depends of the forcing and of the climate sensitivity of the model. This study is also one additional reason to strongly push to continue the development of climate-ice sheet models and to use them more systematically in projections. Overall, I consider thus that this study is an interesting contribution to the topic. However, the authors should be clearer on the limitations of the model they use and I would also suggest to be more quantitative in the evaluation of the feedbacks as detailed below.

We sincerely thank the reviewer for providing thoughtful and constructive comments, especially for suggesting a feedback analysis and more discussions on the model's limitations.

1/ The authors use an Earth system Model of Intermediate Complexity (EMIC), which has a coarser resolution and simplified physics, in particular in the atmosphere, compared to the state-of-the-art General Circulation models (GCM). I have no problem with that as it is currently impossible to perform the number of simulations needed in this study with a GCM. However, the authors must be clearer on this and more explicit on the limitations and uncertainties this brings. First, they should state explicitly when the model is introduced as 'climate model' (line 141 -last paragraph of the introduction) that it is an EMIC since 'climate model' is vague and could be interpreted differently.

Indeed the term “climate model” has been used a bit loosely in the manuscript. UVic-ESCM, as the name suggests, is an Earth system model with reduced complexity especially in its atmosphere module. The model belongs to the category of EMICs, but in this study its biogeochemistry module is turned off with only the atmosphere, ocean, and land components being active. In this light, we prefer to call the model a “climate model of reduced complexity” to avoid giving a misleading impression that it includes “Earth system” processes such as biogeochemical cycles and explicitly-simulated ice sheets. As suggested, we made the following change to the first lines of the last paragraph of the introduction: “Here we provide numerical simulations with a three-dimensional ice sheet model (ISM) quasi-synchronously coupled to a climate model of reduced complexity. The climate model consists of a three-dimensional ocean model, a land model, and a two-dimensional energy-moisture balance model for the atmosphere. Using a reduced-complexity climate model enables carrying out a large set of multi-century scale simulations with different combinations of model

configurations, climate sensitivities, emission scenarios, and initial conditions. Limitations of such model choices will be discussed in the Discussion section.”

The uncertainties should also be made more explicit as the ice sheet climate interactions depend on processes that could not be resolved at the coarse resolution of the ocean and by the (very simplified) atmosphere model applied here. Consequently, the response to freshwater input can be strongly model dependent, high resolution model having in some regions a response to freshwater input of a different sign compared to a low resolution version and thus potentially a different sign for some feedbacks there (e.g., Beadling, et al. 2022). Specifically, the melting in ice sheet cavities is influenced by exchanges at the shelf break, by the circulation within the cavity, by the changes in the salinity on the continental shelf. . . This is represented by a simple quadratic formulation in the model used here. This is probably the best choice at this resolution but the associated limitations should be explained when the representation of ice shelf melting is discussed. If I have understood correctly, meltwater from ice sheet and ice shelf melting is injected at ocean surface but in the real world a large fraction is injected at depth. This could have a strong impact on the oceanic stratification, on the warming at depth and thus on the positive feedback described in the paper (see for instance Mathiot et al. 2017; Jourdain et al. 2017; Mackie et al. 2020). The final paragraph of the manuscript describes somehow those limitations but without details on the processes that may have been missed or key references on the subject, except for the representation of icebergs. This should be expanded in the revised version so that the reader can have a better knowledge of the limitations of this study and of what can be brought in the future when new tools with a better representation of those processes will be available.

Thanks for the thoughtful comment. The original manuscript was indeed lacking when discussing the model’s limitations. We have followed the suggestion and substantially expanded this section. Now it reads:

However, it is important to acknowledge the limitations of this study, and that ice sheet-climate interactions may be strongly model-dependent. The coupled ice sheet-climate model developed for this study was designed as a stopgap until more sophisticated Earth system models with fully integrated ice sheet components become mainstream. Compared with state-of-the-art climate models, the high throughput of the UVic-ESCM comes at the expense of reduced spatial resolution and model complexity, especially for its atmosphere model. Key aspects and processes of the modeling toolkit that need further improvement include but are not limited to:

- *Spatial resolutions: Considerably higher spatial resolutions — in both ocean and atmosphere modules as well as in horizontal and vertical directions — are required to model processes vital for ice shelf basal melting, e.g. incursion of CDW onto continental shelves, and precipitation at ice sheet margins with steep topography. The response to ice sheet meltwater can strongly depend on the ocean model’s spatial resolution and its parameterization schemes for mesoscale eddies and submesoscale eddy restratification. For instance, Antarctic meltwater is more efficiently trapped on the shelf in models with a better resolved and stronger Antarctic Slope Current (ASC), which produces subsurface cooling and suppresses further melt; In contrast, in models with a diffuse ASC,*

ice sheet meltwater more readily escapes to the open ocean, producing strong subsurface warming that accelerates further melt at the base of ice shelves.

- *Ice shelf cavity circulations: The ISM used in this study assumes a simple quadratic relationship between ice shelf basal melt rates and subsurface ocean temperatures at 400-m of nearby ocean cells, without explicitly modeling ocean circulations within the cavity beneath the ice shelf. Intensive basal melting of ice shelves of the Amundsen Sea can induce an overturning circulation in the ice cavity and an inflow of warm water into the cavity, which pumps heat from the deep ocean toward the ocean surface, melting sea ice near the ice sheet margins.*

- *Representation of icebergs: If the MICI mechanism is triggered by intensive warming, collapse of ice shelves and tall, mechanically unstable ice cliffs would release the bulk of FWF in the solid form — i.e. icebergs, which is not explicitly modeled in our study but is treated as added sea ice. Icebergs transport and release freshwater along their tracks from coastal Antarctica to warmer oceans, shifting the regions of freshwater injection equatorward. In addition, most of the large tabular Antarctic icebergs are trapped in counter-clockwise currents along the coast for years before entering the “iceberg alley” of the Weddell Sea and drifting to lower latitudes. This may shift the deposition of icebergs’ freshwater flux westward off the calving sites and lead to more freshwater flux into the Indian Ocean. Iceberg tracks cannot be accurately modeled with the coarse-resolution velocity field from our model, but ocean models with built-in iceberg modules are under active development.*

- *Depth of freshwater injection: The coupling scheme of this study injects ice sheet FWF to the top layer of the ocean model, while in the real world basal meltwater is injected at depth by both ice shelves and icebergs. This modeling choice is due to the lack of explicitly modeled ice shelf cavities and icebergs. Strengthening of ice shelf cavity circulations due to intensified basal melting may deepen the mixed layer around Antarctica, in contrast to iceberg meltwater, which is shown to enhance stratification. Adding iceberg meltwater at depth was found to increase the magnitude of subsurface warming and sea ice trends. This complexity could affect the positive feedback identified in this and previous studies.*

I would also change the last sentence ‘are likely to be confirmed in future work’ because it is impossible to know at this stage how those processes poorly represented in this model affect the conclusions.

The last sentence had probably expressed too much optimism. As suggested, we have replaced it with the following to convey the opinion that further investigations with more complex climate models are needed to fully address ice sheet-climate feedbacks: “*New mechanisms revealed by this coupled ISM-CM model study, including the dependence of ice sheet-climate feedbacks on the pace of warming, and the transition of the net feedback from positive to negative as ice shelves are lost, are potentially important for the sensitivity of ice sheets to climatic warming. They should be investigated more fully in coupled ice sheet-climate models that can better resolve the aforementioned processes that cannot be reliably represented by the model used in this study.*”

2/ The paper deals with ice sheet climate feedbacks but those feedbacks are never really quantified. The difference between experiments is discussed. The sign of the feedback is addressed but there is no precise quantification of the feedback strength. From the experimental design (or with a few additions), it should be possible to quantify the feedbacks using a classical measure such as the feedback factors or the feedback gain, which are widely used for radiative feedbacks but can also be computed for other feedbacks (e.g., Roe 2009, Goosse et al. 2018). This would also provide numbers that can be compared with the results of future studies to determine qualitative differences (i.e. different sign of the feedback) but also quantitatively.

We followed the suggestion by Goosse et al. (2018) on defining a feedback factor and added a paragraph in Methods to describe the feedback analysis for ice sheet FWF-climate interactions:

“The sign and strength of ice sheet FWF-climate feedbacks can be quantified using a feedback factor. The feedback factor γ is defined following Ref.[61]’s recommendation for non-radiative processes:

$$\gamma = \frac{\dot{V}_{iFWF} - \dot{V}_{cFWF}}{\dot{V}_{iFWF}} = 1 - \frac{\dot{V}_{cFWF}}{\dot{V}_{iFWF}}$$

where \dot{V}_{iFWF} is the rate of change in the AIS’ volume in coupled ISM-CM simulations with interactive ice sheet FWF (the total response), \dot{V}_{cFWF} is the rate of change in the AIS’ volume in coupled simulations with constant ice sheet FWF at pre-industrial conditions (the reference response). The perturbation is defined as the change in GMST relative to pre-industrial ($\Delta GMST$). \dot{V}_{iFWF} and \dot{V}_{cFWF} are one-to-one matched based on $\Delta GMST$, so the reference response can be interpreted as the rate of ice loss with the same perturbation in GMST but no interactive ice sheet FWF. In practice, \dot{V}_{iFWF} and \dot{V}_{cFWF} are sampled and matched from bins of GMST with a width of 0.25 K. The centers of GMST bins range from 288 K to 300 K on an interval of 0.25 K. ”

A new figure has been added to the manuscript as Fig. 4, which is also displayed here:

We have added the following paragraph to the end of Section “Competition between positive and negative feedbacks in Antarctica” to discuss the feedback factor and Fig. 4:

“The effect of ice sheet FWF-climate feedbacks on the AIS’ mass loss can be quantified by a feedback factor γ , which is defined as one minus the ratio between the AIS’ mass loss rates in simulations with interactive ice sheet FWF and those in simulations with fixed ice sheet FWF (Method). A positive feedback factor γ indicates that ice sheet FWF-climate feedbacks accelerate the AIS’ mass loss, and vice versa. Fig. 4 presents the feedback factor γ as a function of the rate of ice loss from Antarctica (in Sv, freshwater flux equivalent). γ is positive for low ice loss rates and transitions to negative values around a threshold of 0.2 Sv. The transition of the net feedback from positive to negative is consistent with Fig. 2. Peak amplitude of γ exceeds 0.5 for both the positive and the negative feedback regimes, indicating moderately strong feedbacks for ice sheet FWF-climate interactions. The strength of the negative feedback decreases as the ice loss rate exceeds ~ 0.4 Sv, corresponding to the stage of a WAIS collapse. In this stage, ice sheet instability mechanisms are at work, and the rate of ice loss is more strongly affected by ice sheet dynamics than atmospheric

Fig. 4 Ice loss-freshwater flux feedback factor as a function of ice loss rate from the AIS. Blue markers are based on simulations under the historical-SSP2-4.5 scenario with an ECS of 4.0°C, while red markers are based on simulations under the historical-SSP5-8.5 scenario with an ECS of 5.6°C. Definition of the feedback factor and its calculation are described in Methods “Feedback analysis”.

and oceanic thermal forcings, which may explain the decreasing feedback strength and irregularities at very large ice loss rates.”

Specific points

Line 218. It is not clear for me from the main text nor from the appendix how the ‘fixed ice sheet FWF’ are computed and how they compare to the ones computed interactively (Fig 1ab). Is it the FWF fluxes from the climate model only? Are they derived from an estimate for present-day conditions?

In simulations with “fixed ice sheet FWF”, FW fluxes are derived from coupled ISM-CM simulations under pre-industrial conditions. Under *historical* and future scenarios, FW fluxes injected to the ocean are kept fixed at the pre-industrial level, and the FW fluxes computed by the ISM are not used by the climate model. To improve clarity, we have added a sentence to the beginning of this paragraph: “A group of simulations are carried out with “fixed ice sheet FWF”, i.e. ice sheet FWF received by the climate model is kept same as the long-term mean FWF simulated by the coupled ISM-climate model in pre-industrial conditions.”

The tar files I uploaded includes movies but I have not seen any reference or explanation for those movies.

In the main text, we have added references for movies 1 and 2 in the caption of Fig. 3 and for movie 3 in the caption of Extended Data Fig. 14.

References

- Beadling, et al. 2022. Importance of the Antarctic Slope Current in the Southern Ocean response to ice sheet melt and wind stress change. Journal of Geophysical Research: Oceans, 127, e2021JC017608. <https://doi.org/10.1029/2021JC017608>*
- Goosse H et al., 2018. Quantifying climate feedbacks in polar regions. Nature Communications 9, 1919, DOI: 10.1038/s41467-018-04173-0*
- Jourdain, et al. 2017. Ocean circulation and sea-ice thinning induced by melting ice shelves in the Amundsen Sea, J. Geophys. Res. Oceans, 122, 2550–2573, doi:10.1002/2016JC012509.*
- Mackie et al. 2020. Climate Response to Increasing Antarctic Iceberg and Ice Shelf Melt. Journal of Climate 33, 8917 DOI: 10.1175/JCLI-D-19-0881.1*
- Mathiot et al. 2017. Explicit representation and parametrised impacts of under ice shelf seas in the zcoordinate ocean model NEMO 3.6. Geosci. Model Dev., 10 (7) (2017), p. 2849*
- Roe, G. H., 2009 Feedbacks, timescales and seeing red. Annu. Rev. Earth. Planet. Sci. 37, 93–115. <https://doi.org/10.1146/annurev.earth.061008.134734>.*

These references have been added to the revised manuscript.

Reviewer #3

General This study uses an asynchronously (once-per-year) coupled climate-ice sheet model to quantify the feedbacks of freshwater release from the ice sheets on future ice sheet mass loss. The processes that are most relevant and under investigation here are enhanced stratification of the upper ocean, cooling the upper ocean and atmosphere (negative feedback), and warming the deeper ocean layers (positive feedback). Then there is the weakening of the AMOC, reducing interhemispheric heat exchange by the ocean (bipolar see-saw). The technical quality of the paper is good: it is well-written with high-quality figures, and the results and acting processes are clearly explained. My main concern is that the adopted highly idealized/simplified parametrizations in the low-resolution climate model could lead to very large uncertainties in the modeled ice sheet mass balance, to the point that I question whether the approach is sufficiently robust to answer the main research questions. The authors are aware of these potential limitations and rightfully point out that it will take many years before full-fledged, high-resolution ESMs can be run for such extended periods.

We sincerely thank the reviewer for providing thoughtful and constructive comments, especially for suggesting a model evaluation so that readers could better grasp the model's performance as well as limitations.

Major comments

I acknowledge the need for fast climate models for these applications involving ice sheet-ocean interactions, and I appreciate the efforts of the authors to (asynchronously) couple climate and ice sheet models. However, my main concern after reading this manuscript is that the climate model used is so coarse (order several 100s km horizontal resolution), and its forcings and parametrizations to calculate moisture and heat transport in ocean, atmosphere, ice sheets, and the interfaces between them so simplified (e.g., basal and surface melt) that it is unclear whether it is at all possible to robustly model ice sheet climate interactions (see description of the Climate Model line 898 and further). Will this model setup really resolve the "contradicting conclusions" (l. 120) of previous work done?

These are legitimate concerns. We acknowledge that the climate model has a very simple atmosphere component (a two-dimensional energy and moisture balance model), so processes important for transports of heat and moisture in the real atmosphere, e.g. the Hadley Circulation and mid-latitude eddies, cannot be reliably modeled here. The climate model also suffers a too weak polar amplification, as shown in Extended Data Fig. 3 and 4, so we have to use an emulated ECS of 4.0°C to generate about the same degree of polar warming as a comprehensive climate model with an ECS of 3.0°C (IPCC-AR6 best estimate) would do. These limitations, along with several key processes that are not well-represented by this model – but potentially important for ice sheet-climate interactions – are now discussed more fully in the Discussion section of the revised manuscript.

Despite the limitations, the coupled ISM-CM simulates both the positive feedback and the negative

feedback in a self-consistent way. The climate model in this study has a complexity on par with LOVECLIM used by Golledge et al. (2019) [G19], the ice sheet model (PSUICE3D) is the same one used by DeConto et al. (2021) [D21] and Sadai et al. (2020) [S20], and is similar in complexity to PISM (used in G19). The interfaces of ISM and CM in previous studies are of similar complexity as those in our study, but our short coupling step (1 year) provides near-synchronous coupling between the ISM and the CM, which in theory offers a more reliable diagnosis of ice sheet-climate feedbacks than the offline coupling approach in previous studies. The new Fig. 4 shows that the feedback factor transitions from positive to negative values when the ice loss rate exceeds ~ 0.2 Sv (freshwater flux equivalent). As explained in the first paragraph of the Discussion section, the higher ECS in CCSM4 (4.0 °C, compared to LOVECLIM's 1.9 °C) and the MICI mechanism lead to faster AIS retreat and higher FWF, which is consistent with the predominantly negative feedback identified in S20 and D21. Based on these findings, we would suggest the coupled ISM-CM modeling in our study reconciles contradicting conclusions drawn by previous studies.

The authors claim that (l. 494): "... first order features identified from this study, including the inter-hemispheric interaction between retreating ice sheets, the competing positive and negative feedbacks, the dependence of these feedbacks on the pace of warming, and positive feedback diminishing as ice shelves are lost, are likely to be confirmed in future work." But there is no way to really tell.

In the revised manuscript, we have replaced this statement with the following to convey the opinion that further investigations with more complex climate models are needed to study ice sheet-climate feedbacks: *"New mechanisms revealed by this coupled ISM-CM model study, including the dependence of ice sheet-climate feedbacks on the pace of warming, and the transition of the net feedback from positive to negative as ice shelves are lost, are potentially important for the sensitivity of ice sheets to climatic warming. They should be investigated more fully in coupled ice sheet-climate models that can better resolve the aforementioned processes that cannot be reliably represented by the model used in this study."*

A concrete, bare minimum improvement of the manuscript should include a (spatially distributed, not spatially integrated) model evaluation to demonstrate that the contemporary mass balance of the ice sheets is well captured, as well as their first-order sensitivity to atmospheric (Greenland) and oceanic (Antarctica) warming, as observed in the last decades. This includes a separate evaluation of the main mass balance components, i.e., surface mass balance (snowfall, runoff), ice discharge (ice velocity, ice thickness, basal melt of Antarctic ice shelves), and their recent trends that invoked the contemporary mass loss of both ice sheets.

We agree that some model evaluation would let readers have a more balanced impression on the study's strength and limitations. We followed the reviewer's suggestion to present model evaluation for both ice sheets' mass balance components and their recent trends. We also compare mass anomalies of both ice sheets as estimated by IMBIE 2021 with those simulated by the coupled ISM-CM under the historical-SSP2-4.5 scenario with an ECS of 4.0°C. These figures are attached below, and they have also been included in the revised Extended Data as Fig. 18-22. Methods section "Ice sheet-climate coupling" has been updated accordingly.

Please note that these new figures are more of “sanity check” rather than “model validation”. We should not expect the relatively simple climate model (UVic-ESCM) to reproduce the transient climate fields that lead to observed trends in each ice sheet’s surface mass balance. Trends in both ice sheets’ SMB and the AIS’ basal melt are strongly influenced by decadal and multi-decadal climate variabilities, which nonetheless are absent in UVic-ESCM. Without proper initiation procedures, even a comprehensive climate model cannot reproduce the observed climate trends on decadal and multi-decadal time scales. In addition, under historical-SSP scenarios, the coupled ISM-CM is forced by changes in atmospheric CO₂ concentration only, ignoring other forcing agents such as methane, aerosols, and changes in land usage. UVic-ESCM displays a too weak polar amplification, which has been (to some degree) compensated by using an ECS (4.0°C) higher than the best-estimate for IPCC-AR6 (3.0°C).

Despite low expectations, the coupled ISM-CM simulates reasonable good SMB components for the GrIS, and a trend in the ice sheet’s mass similar to that estimated by IMBIE 2021 (Extended Data Fig. 18, 20). This suggests the coupled model has an acceptable ice sheet sensitivity to greenhouse gas forcing. When forced yearly by reanalysis climate fields (CERA-20C, available from 1901-2010), the time series of GrIS mass anomaly simulated by the ISM more closely matches the IMBIE data(Extended Data Fig. 19, 20), indicating the sensitivity of GrIS’ mass balance to climate forcing is well-captured by the ISM.

For the AIS, the coupled ISM-CM has a less optimum performance in modeling the ice sheet’s contemporary mass balance (Extended Data Fig. 21). In particular, the coupled ISM-CM simulates too extensive ice shelves and too little surface melt under modern climate conditions. The modeled ice loss is generally slower than that estimated by IMBIE 2021 (Extended Data Fig. 22). In addition to UVic-ESCM’s own limitations, decadal and multi-decadal climate variabilities over recent decades, which cannot be reproduced by the coupled ISM-CM, may partially explain the differences.

Although the mass balance simulated by the coupled ISM-CM may be subject to significant bias, and the simulated ice loss may be somewhat delayed (Extended Data Fig. 22), these issues are not essential for the main research objectives of this study. This modeling study is not meant to give accurate, time-specific future projections of ice sheets, but to explore various combinations of emission scenarios, climate sensitivities, and options for ice sheet FWF to find patterns and regularities of ice sheet-climate interactions via freshwater flux. Main conclusions drawn from this study, including the presence of competing positive and negative feedbacks, transition of the AIS’ net feedback from positive to negative as FWF exceeds a certain threshold, and retreating ice sheets interacts via changing the AMOC, are not likely to be strongly affected by the climate model’s bias and lack of complexity.

In the introduction, frequent reference is made to glacial-interglacial cycles and the associated meltwater pulses. But today’s climate and ice volume are very different from those during glacial epochs (higher temperatures, smaller ice volumes), and the expected response to meltwater pulses is therefore (very) different. Please comment on whether/why these comparisons are still warranted.

Extended Data Fig. 18 Top row: State of the Greenland Ice Sheet in year 2020 as simulated by the coupled ISM-CM under historical-SSP2-4.5 scenario with an ECS of 4.0°C (10-member ensemble mean); Bottom row: Change in each variable between year 2020 and year 1990.

Extended Data Fig. 19 Top row: State of the Greenland Ice Sheet as simulated by the PSUICE3D ice sheet model forced by yearly climate fields from CERA20C; Bottom row: Difference in each variable between 1990s and 2000s.

Extended Data Fig. 20 Anomaly in the Greenland Ice Sheet’s mass as estimated in IMBIE 2021 and that simulated by the coupled ISM-CM under historical-SSP2-4.5 scenario with an ECS of 4.0°C.

Extended Data Fig. 21 Top row: State of the Antarctic Ice Sheet in year 2020 as simulated by the coupled ISM-CM under historical-SSP2-4.5 scenario with an ECS of 4.0°C (10-member ensemble mean); Bottom row: Change in each variable between year 2020 and year 1990.

Extended Data Fig. 22 Anomaly in the Antarctic Ice Sheet’s mass as estimated in IMBIE 2021 and that simulated by the coupled ISM-CM under historical-SSP2-4.5 scenario with an ECS of 4.0°C.

Reference to the climate effects of ice sheet FWF over glacial-interglacial cycles were meant to give readers a historical background for research on ice sheet-climate interactions, and to suggest that large freshwater discharge events during deglaciations could have substantial regional and global climate impacts. We do share the reviewer’s view that present-day climate and ice volume are drastically different from those epochs, and climate perturbations during past meltwater pulses are not analogs of what would happen in future anthropogenic warming scenarios. To clarify this view, we have added the following to the second paragraph of Introduction: *“Nowadays ice sheet geometry and discharge locations of ice sheet meltwater are very different from those over past glacial-interglacial cycles. Climate perturbations caused by past meltwater pulses, therefore, should not be regarded as direct analogs of what would happen in intensive future warming scenarios, prompting the need for modeling the climate effects of future ice sheet melt.”*

Fig. 1: Can you explain is the AIS response so much less regular than the GrIS response? From the paleo-record, we know that the GrIS climate is very sensitive to changes in the AMOC strength.

We are sorry if we had understood this question incorrectly, specifically whether “response” refers to sea level contribution or FWF. In Fig. 1, time series of sea level contribution for the AIS do not look more irregular than those for the GrIS. Time series of Antarctic FWF indeed show more “wiggles”, but that is probably due to large-scale ice shelf calving/collapse events or phases of fast grounding line retreat. UVic-ESCM does not show a multi-decadal oscillation of the AMOC or an Atlantic Multi-decadal Oscillation (AMO), neither does the model display an ENSO. Due to

the lack of these internal climate variabilities, climate over Greenland changes relatively smoothly under greenhouse gas forcings. The AMOC, however, presents a weak centennial variability in our coupled ISM-CM with interactive AIS FWF, which is likely a result of interactions between basal melting of ice shelves and the AMOC. This is the cause for the significantly larger spread in the AIS' sea level contribution (compared with the GrIS) between ensemble members in the bottom two rows of Fig. 2. We have not thoroughly investigated the nature and mechanisms of this centennial AMOC variability, because the climate model used here is probably too simplified for that task. The presence of centennial variability makes it necessary to do ensemble simulations to robustly isolate the feedback signal of Antarctic FWF, especially in the early stage of warming (see insets in Fig. 2 that zoom-in to the period before 2100). In the current study, we use the 10-member ensemble-mean to remove the influence of this AMOC centennial variability. But for now we leave the nature of this variability for future investigations using a more comprehensive climate model.

Minor and textual comments

The first line of the abstract: the list of processes that influence freshwater fluxes from ice sheets is incomplete and appears rather arbitrary. I suggest only mentioning the feedbacks central to this paper.

The first line of the abstract was indeed ambiguous. What we meant to say was like: “Anthropogenic warming increases freshwater fluxes from ice sheets, which would change ocean stratification, atmosphere-ocean heat exchange, and ocean circulations.” So ocean stratification, atmosphere-ocean heat exchange, and ocean circulations are what to be *influenced by* ice sheet freshwater fluxes. However, the sentence could also be interpreted as “The increase in ice sheet freshwater fluxes is due to anthropogenic warming *and* changes in ocean stratification and so on.” To avoid such ambiguity, we have revised the first sentence to: “*Under anthropogenic warming, freshwater fluxes from ice sheets are projected to increase in the coming centuries, changing ocean stratification, atmosphere-ocean heat exchange, and ocean circulation.*”

l. 113: "Surface cooling reduces ice surface melting and ice shelf crevassing and calving..." The impact of atmospheric cooling on ice shelf crevassing seems limited to me. Are the authors referring here to the hydrofracturing of ice shelves following meltwater ponding?

Yes, here we were referring to the hydrofracturing process. In the revised manuscript, the sentence has been changed to the following to improve clarity: “*Surface cooling reduces ice surface melting and meltwater-induced hydrofracturing and calving of ice shelves ...*”

l. 131: "... in the existing offline coupling framework..." Currently, various fully coupled models are being tested and/or run (e.g., UKESM). Please mention these.

We have added the following sentence to clarify that fully coupled climate-ice sheet models are under development but have not been used for studying feedbacks on centennial time scales: “*Although various Earth system models with built-in ice sheet components are under active development, e.g.*

the UKESM and the E3SM, they have not been used for studying centennial-scale ice sheet-climate feedbacks.”

REVIEWERS' COMMENTS

Reviewer #1 (Remarks to the Author):

I feel that the manuscript is nearly ready to be published, and that the authors have satisfactorily addressed the concerns of the reviewers. The only somewhat major point is that the Extended Data Fig. 14 appears to conflict with Figure 3 and some of the discussion in the manuscript.

Specific comments:

1. Line 041: Change "ocean circulations" to "ocean circulation".
2. Line 055: For clarity add "freshwater discharge from one ice sheet" before "exacerbates the opposing ice sheet's..."
3. Line 142: Add "(discussed further in the methods section)" after "offline-coupled ice sheet models" to direct the reader to further details.
4. Line 200: Suggest adding "and their potential contribution to sea level is recorded" after "ice sheets respond to changes in climate". I was still a bit confused by "sea level contribution" mentioned in the figures and this would help in clarifying this.
5. Line 218: Change "future warming" to "future global warming".
6. Line 263: Add "(Fig. 1 a,b)" after "between scenarios".
7. Line 283: Again, suggest adding "but potential ice sheet sea level contribution is recorded" at the end of this sentence for clarity.
8. Line 309: Suggest changing "when its FWF" to "when FWF" as the text here is still referring to simulations where FWF is interactive from both ice sheets.
9. Line 370: A reference to "(Fig. 2 l)" can be added here.
10. Line 377: Add "(Fig. 3 b,d)" after "subsurface warming".
11. Line 409: Add "(Fig. 2 n)" after "early 2100s."
12. Line 420: Change "Method" to "Methods".
13. Line 428: Add "The" before "Peak amplitude".
14. Line 433: Change "stage of a" to "stage of the".
15. Lines 471-491: I believe Extended Data Figures 14-16 have not been referred to in the text. This might be a good place to do so.
16. Line 515: Change "effective ECS" to "ECSs".
17. Lines 583-585: Change "subsurface ocean temperatures at 400 m" to "400 m depth ocean temperatures"
18. Line 707: Add "a" before "marine ice sheet".
19. Line 790-791: Change " ρ_w is the density of ice" to " ρ_i is the density of ice"
20. Line 861: The figure only shows various curves for the model. Can values for the modeled and observed volumes be specified here?
21. Line 853: Clarify if this is the top of atmosphere or surface flux.
22. Line 878: Change "displays" to "display".
23. Line 965: Change "changes" to "change".
24. Line 977: Change "are 1.07" to "is 1.07".
25. Line 1101: Change to read "the ice-albedo feedback associated with the sea ice surface"
26. Line 1111: Change "combined as a" to "combined as an"
27. Line 1141: Change "variabilities" to "variability".
28. Line 1143: Change "responsible the differences" to "responsible for the differences"
29. Line 1166: Add "The" before "ice-albedo feedback".
30. Lines 1168-1170: Change to read "...are naturally resolved in the coupled model. They are not..."
31. Line 1203: Begin a new sentence with "Each suite..."
32. Line 1217: Change "natural variabilities" to "natural variability".
33. Extended Data Figure 3: Specify units for surface air temperature.
34. Extended Data Figure 5: If possible could the figure be modified so that the legend does not overlap with the lines that are being plotted? (e.g. by extending the x-axis range.)
35. Extended Data Figure 8: Change to read "... (FWAG, solid lines) for coupled model simulations..."

Also mention that a), b), and c) show snapshots at the years 2000, 2300 and 2500 respectively.

36. Extended Data Figure 14:

37. Extended Data Figures 14-16, and video 3: The results here do not appear to agree with Figure 3, or discussion in the text that in the SSP5-8.5 scenario with ECS = 5.6 °C, there is cooling at the surface near Antarctica by 2300. Is it possible that these figures have been accidentally mislabeled, and that these figures are for SSP2-4.5 with ECS=4.0 °C? If so these figures should be updated or the captions changed.

38. Extended Data Figure 17: The legend for the different colors appears to be missing here.

Reviewer #2 (Remarks to the Author):

The authors have well addressed the comments of the reviewers, in particular mine. I thus consider that the manuscript is ready for publication.

I have listed a few minor suggestions below that the authors may want or not to include in their final version.

1/ The abstract could be more precise. For instance, it can be interesting to mention at which level the feedback shifts from positive to negative or to quantify the magnitude of positive and negative feedbacks.

2/ Line 401-403. Would it be possible to quantify this estimate. For instance, does the negative feedback dominates when all the ice shelves have collapsed or earlier than that?

3/ Fig. 4. Is the value of the feedback parameter similar for Greenland and Antarctic ice sheets ?

4/ Line 550. Maybe it would be useful to mention that the present estimate is the first one and gives thus a benchmark that can be used for comparison in subsequent studies.

5/ Line 561. Here maybe a way to discuss the limitations is to state that the model represents the large-scale processes controlling the feedbacks, but not the small-scale ones.

6/ Line 1046-1047. Does it mean that the coupled system is losing energy as this latent heat is not passed to the CM ?

7/ Line 1143. Change to 'responsible for the difference'.

Reviewer #3 (Remarks to the Author):

Thank you to the authors for their extensive answer to reviewer comments and the associated revisions. I think the authors did a good job in addressing my comments, and I am happy to recommend publication of the MS.

NCOMMS-23-27716B

Competing climate feedbacks of ice sheet freshwater discharge in a warming world

RESPONSE TO REVIEWS

We would like to thank the reviewers again for providing constructive comments for further improvement of our manuscript. The manuscript has been revised accordingly. Please find the point-by-point responses in the following pages, where the reviewers' comments are in *blue italic*, and our responses in black text.

Dawei Li, On behalf of all co-authors,

May 26, 2024

Reviewer #1

I feel that the manuscript is nearly ready to be published, and that the authors have satisfactorily addressed the concerns of the reviewers. The only somewhat major point is that the Extended Data Fig. 14 appears to conflict with Figure 3 and some of the discussion in the manuscript.

We sincerely thank the reviewer for providing additional, very detailed comments, which are of much help in improving the clarity of the manuscript. We ensure that Supplementary Fig. 14 is consistent with Figure 3 and relative discussion, as explained later in response to specific comment #36, 37.

Specific comments: 1. Line 041: Change “ocean circulations” to “ocean circulation”.

Done.

2. Line 055: For clarity add “freshwater discharge from one ice sheet” before “exacerbates the opposing ice sheet’s...”

Done.

3. Line 142: Add “(discussed further in the methods section)” after “offline-coupled ice sheet models” to direct the reader to further details.

Done.

4. Line 200: Suggest adding “and their potential contribution to sea level is recorded” after “ice sheets respond to changes in climate”. I was still a bit confused by “sea level contribution” mentioned in the figures and this would help in clarifying this.

Done.

5. Line 218: Change “future warming” to “future global warming”.

Done.

6. Line 263: Add “(Fig. 1 a,b)” after “between scenarios”.

Done.

7. Line 283: Again, suggest adding “but potential ice sheet sea level contribution is recorded” at the end of this sentence for clarity.

Done.

8. Line 309: Suggest changing “when its FWF” to “when FWF” as the text here is still referring to simulations where FWF is interactive from both ice sheets.

Done.

9. Line 370: A reference to “(Fig. 2 l)” can be added here.

Done.

10. Line 377: Add “(Fig. 3 b,d)” after “subsurface warming”.

Done.

11. Line 409: Add “(Fig. 2 n)” after “early 2100s.”

Done.

12. Line 420: Change “Method” to “Methods”.

Done.

13. Line 428: Add “The” before “Peak amplitude”.

Done.

14. Line 433: Change “stage of a” to “stage of the”.

Done.

15. Lines 471-491: I believe Extended Data Figures 14-16 have not been referred to in the text. This might be a good place to do so.

Extended Data Figures 14-16 (now Supplementary Fig. 14-16) are referred to as “This reduces the density of AAIW, resulting in a stronger inflow of NADW into the SH and strengthening the AMOC, which warms the North Atlantic and enhances ice loss from the GIS (Supplementary Fig. 9, 11, 12, 14, 15, 16)”.

16. Line 515: Change “effective ECS” to “ECSs”.

Done.

17. Lines 583-585: Change “subsurface ocean temperatures at 400 m” to “400 m depth ocean temperatures”

Done. We also changed “subsurface ocean temperatures at 400 m” to “400 m depth ocean temper-

atures” in the caption of Fig. 3.

18. Line 707: Add “a” before “marine ice sheet”.

Done.

19. Line 790-791: Change “ ρ_w is the density of ice” to “ ρ_i is the density of ice”

Done.

20. Line 861: The figure only shows various curves for the model. Can values for the modeled and observed volumes be specified here?

Line 816? The line has been changed to while TPDD = -4.0°C results in a modeled near-equilibrium GIS volume close to modern observation (7.5 m versus 7.4 m SLE, Supplementary Fig. 5).

21. Line 853: Clarify if this is the top of atmosphere or surface flux.

We have changed the phrase to “the outgoing infrared flux at the top of the atmosphere”

22. Line 878: Change “displays” to “display”.

Done.

23. Line 965: Change “changes” to “change”.

Done.

24. Line 977: Change “are 1.07” to “is 1.07”.

Done.

25. Line 1101: Change to read “the ice-albedo feedback associated with the sea ice surface”

Done.

26. Line 1111: Change “combined as a” to “combined as an”

Done.

27. Line 1141: Change “variabilities” to “variability”.

Done.

28. Line 1143: Change “responsible the differences” to “responsible for the differences”

Done.

29. *Line 1166: Add “The” before “ice-albedo feedback”.*

Done.

30. *Lines 1168-1170: Change to read “...are naturally resolved in the coupled model. They are not...”*

Done.

31. *Line 1203: Begin a new sentence with “Each suite...”*

Done.

32. *Line 1217: Change “natural variabilities” to “natural variability”.*

Done.

33. *Extended Data Figure 3: Specify units for surface air temperature.*

Extended Data Figure 3 (now Supplementary Fig. 3) shows the amplification factor, which is dimensionless. To clarify its definition, we have modified the caption to read “The (dimensionless) amplification factor is defined as the change in ...”.

34. *Extended Data Figure 5: If possible could the figure be modified so that the legend does not overlap with the lines that are being plotted? (e.g. by extending the x-axis range.)*

For each panel of Supplementary Fig. 5, we have extended the x-axis range so that the legend stays clear of the lines.

35. *Extended Data Figure 8: Change to read “... (FWAG, solid lines) for coupled model simulations...” Also mention that a), b), and c) show snapshots at the years 2000, 2300 and 2500 respectively.*

Done.

36. *Extended Data Figure 14: 37. Extended Data Figures 14-16, and video 3: The results here do not appear to agree with Figure 3, or discussion in the text that in the SSP5-8.5 scenario with ECS = 5.6 °C, there is cooling at the surface near Antarctica by 2300. Is it possible that these figures have been accidentally mislabeled, and that these figures are for SSP2-4.5 with ECS=4.0 °C? If so these figures should be updated or the captions changed.*

Extended Data Figures 14-16 (now Supplementary Fig. 14-16), and Supplementary Video 3 are consistent with Figure 3 and associated discussions. They actually show surface cooling near Antarctica by 2300, but the cooling is restricted to the top 50-100 m – a thin veneer that can be easily overlooked. For instance, here FIG. 1 shows the lower-left panel of Supplementary Fig. 14 zoomed into the top 500 m, which clearly displays surface cooling near Antarctica in the fourth

row (dT).

FIG. 1: Snapshots of latitude-depth cross-sections of selected variables in year 2300 from coupled ISM-CM simulation with the FWAG (interactive FWF for both ice sheets) configuration under historical-SSP5-8.5 and ECS = 5.6 °C. This is the same as the lower-left panel of Supplementary Fig. 14 but zooms into the top 500 m.

38. Extended Data Figure 17: The legend for the different colors appears to be missing here.

We have added a legend for this figure (now Supplementary Fig. 17).

Reviewer #2

The authors have well addressed the comments of the reviewers, in particular mine. I thus consider that the manuscript is ready for publication. I have listed a few minor suggestions below that the authors may want or not to include in their final version.

We thank the reviewer for these additional suggestions for a further improvement. The manuscript has been modified accordingly.

1/ The abstract could be more precise. For instance, it can be interesting to mention at which level the feedback shifts from positive to negative or to quantify the magnitude of positive and negative feedbacks.

Thanks for the suggestion but we have to shorten the abstract to fewer than 150 words to meet the journal's requirement. There is simply no space for describing the findings more quantitatively.

2/ Line 401-403. Would it be possible to quantify this estimate. For instance, does the negative feedback dominates when all the ice shelves have collapsed or earlier than that?

We have tried and plotted the feedback factor against the total area of Antarctic ice shelves, but there appeared to be no obvious one-to-one relationship. Fig. 4 shows that the feedback factor strongly depends on the ice loss rate, which in turn depends on the area of ice shelves and the degree of oceanic warming. This may explain the absence of a definitive threshold of ice shelf area for the overall feedback to transition from positive to negative.

3/ Fig. 4. Is the value of the feedback parameter similar for Greenland and Antarctic ice sheets ?

The interactions between the Greenland Ice Sheet and the climate via ice sheet freshwater flux only display a negative feedback, due to the lack of extensive ice shelves. Therefore, the feedback analysis was only done for the Antarctic Ice Sheet.

4/ Line 550. Maybe it would be useful to mention that the present estimate is the first one and gives thus a benchmark that can be used for comparison in subsequent studies.

A sentence has been added here: "It provides a benchmark that can be used for comparison in subsequent studies using more comprehensive models."

5/ Line 561. Here maybe a way to discuss the limitations is to state that the model represents the large-scale processes controlling the feedbacks, but not the small-scale ones.

As suggested, we have added the following sentence here: "The coupled ice sheet-climate model represents the large-scale processes controlling the feedbacks, but not the small-scale ones."

6/ Line 1046-1047. Does it mean that the coupled system is losing energy as this latent heat is not passed to the CM ?

This is the latent heat required for melting the ice (top) surface, so its omission means that the atmosphere is not losing as much energy as it should be. This should result in a slight warm bias, but not so much as to significantly change the responses in atmospheric and oceanic temperatures to ice sheet freshwater flux.

7/ Line 1143. Change to 'responsible for the difference'.

Done.

Reviewer #3

Thank you to the authors for their extensive answer to reviewer comments and the associated revisions. I think the authors did a good job in addressing my comments, and I am happy to recommend publication of the MS.

We would like to thank the reviewer again for providing very constructive comments, which helped much in improving the manuscript.